# Be Confident! Towards Trustworthy Graph Neural Networks via Confidence Calibration

**Xiao Wang, Hongrui Liu, Chuan Shi,** *Cheng Yang
School of Computer Science (National Pilot Software Engineering School)
Beijing University of Posts and Telecommunications
Beijing, China
{xiaowang, liuhongrui, shichuan, yangcheng}@bupt.edu.cn

## Abstract

Despite Graph Neural Networks (GNNs) have achieved remarkable accuracy, whether the results are trustworthy is still unexplored. Previous studies suggest that many modern neural networks are over-confident on the predictions, however, surprisingly, we discover that GNNs are primarily in the opposite direction, i.e., GNNs are under-confident. Therefore, the confidence calibration for GNNs is highly desired. In this paper, we propose a novel trustworthy GNN model by designing a topology-aware post-hoc calibration function. Specifically, we first verify that the confidence distribution in a graph has homophily property, and this finding inspires us to design a calibration GNN model (CaGCN) to learn the calibration function. CaGCN is able to obtain a unique transformation from logits of GNNs to the calibrated confidence for each node, meanwhile, such transformation is able to preserve the order between classes, satisfying the accuracy-preserving property. Moreover, we apply the calibration GNN to self-training framework, showing that more trustworthy pseudo labels can be obtained with the calibrated confidence and further improve the performance. Extensive experiments demonstrate the effectiveness of our proposed model in terms of both calibration and accuracy.

## 1 Introduction

Graphs are ubiquitous in the real world, including social networks, e-commerce networks, traffic networks, and so on. Recently, Graph Neural Networks (GNNs), which are able to effectively learn the node representations based on the message-passing manner, have attracted considerable attention in dealing with graph data [16, 33, 39, 44, 15, 2, 34]. To date, GNNs have been applied to various applications and achieved remarkable accuracy, e.g., node classification [16, 33], link prediction [41] and graph classification [9].

However, it is well established that a model with good accuracy is not the only goal, but a trustworthy model is highly desired in many applications, especially in safety-critical fields [1]. Usually, a trustworthy model implies that it should know when it is likely to be incorrect, in other words, the probability, i.e., the confidence, associated with the predicted class label should reflect its ground truth correctness likelihood [12]. For example, in the scene of autonomous driving, the system will adopt the prediction given by the model only when the model has high confidence for its prediction. Otherwise, the decision-making power will be returned to the driver or the system adopts other safer strategies. Recently, the confidence calibration has attracted considerable attention in deep learning [12, 40, 19], which reveals that many modern neural network models are over-confident on the predictions, i.e., the prediction accuracy is lower than its confidence. However, it has not been studied

---

*Corresponding author

35th Conference on Neural Information Processing Systems (NeurIPS 2021).

in GNNs on the semi-supervised scenario, which gives rise to one fundamental question: *will the current GNNs follow the same over-confident property as other neural networks?* A well-informed answer can help us better understand GNNs and enable GNNs to be applied to various areas in a more reliable manner.

As the first contribution of this study, we present experiments assessing the relationship between the confidence and the accuracy of Graph Convolutional Networks (GCNs) [16] and Graph Attention Networks (GAT) [33] in the node classification task (more details can be seen in Section 2), respectively. Surprisingly, we discover that existing GNNs are far distant from being well-calibrated, and more importantly, GNNs tend to be under-confident in their predictions, which is very different from other modern deep learning models that are often over-confident [12, 19]. GNNs being under-confident means that many predictions are distributed in the low-confidence range, and therefore, fewer predictions are available for safety-critical applications. Once the weakness is identified, another natural question is: *how can we calibrate the confidence on predictions given by GNNs so as to make them more trustworthy?*

Essentially, the confidence calibration is to calibrate the outputs (also known logits) of original models (e.g., GNNs), therefore, a straightforward manner is to employ temperature scaling (TS) [12], OP-families [25] to learn calibration function using a held-out dataset in a post-hoc way. However, when being applied to graphs, they all ignore the effect of topology, which will inevitably make mistakes during calibration. For example, considering that the logits of two nodes *a* and *b* are the same, but node *a* is similar to its neighbors while node *b* is not. Apparently, the predictions of GCNs for *a* should be more confident than *b*, while the traditional calibration methods, e.g., TS, will learn the same confidence for *a* and *b*, because it does not consider the effect of topology.

Moreover, most of them explore calibration functions only in the linear space [12, 18] while it is well known that non-linear space contains more complex function transformation which is able to calibrate networks with complicated landscapes well. Even if some works have explored the non-linear space such as Matrix Scaling [12], they generally degrade the classification accuracy of the original classifier, while a good accuracy is still a basic requirement by many applications.

In this paper, we introduce a topology-aware post-hoc calibration method for GNNs. Specifically, for the logits given by the original classification GNNs, we employ another calibration GCN (CaGCN) to propagate confidence, naturally enabling that the confidence of topologically adjacent nodes becomes similar. CaGCN learns a unique temperature *t* for each node for temperature scaling, thus preserving the accuracy of the original classification GCN. In addition, based on our finding that large numbers of high-accuracy predictions are distributed in the low-confidence range, we design a calibrated self-training model CaGCN-st in which the confidence is firstly calibrated then used to generate pseudo labels with high confidence. The contributions of this paper are three-fold:

- We study the trustworthy problem of GNNs, and discover one unique characteristic of GNNs, i.e., the predictions made by GNNs are usually under-confident.
- We propose a novel trustworthy GNN model based on the confidence calibration. Our proposed calibration function has three features: topology-aware, non-linear, and accuracy-preserving. We further design a calibrated self-training GNN model, which can effectively utilize the predictions with high confidence.
- Extensive experiments demonstrate the effectiveness of our proposed models in terms of both calibration and accuracy.

## 2 Notation and Preliminary Study

In this paper, we focus on the calibration of semi-supervised node classification in an undirected attributed graph $G = (V, E)$ with the adjacent matrix $\mathbf{A} \in \mathbb{R}^{N \times N}$ and the node feature matrix $\mathbf{X} = [\mathbf{x}_1, \ldots, \mathbf{x}_N]^{\mathsf{T}}$. $V$ is a set of nodes and $E \subseteq V \times V$ is a set of edges between nodes. $N = |V|$ is the number of nodes. Here we give the definition of perfect calibration of GNNs as follows:

**Definition 1.** *Given random variables* $\mathbf{A}$, $\mathbf{X}$, $\mathbf{Y} \subseteq \{1, \ldots, K\}$ *and a GNN model* $f_\theta$ *where* $\theta$ *is the learnable parameters, for node i with label* $y_i \in \mathbf{Y}$, $\mathbf{z}_i = f_\theta(\mathbf{x}_i, \mathbf{A}) = [z_{i,1}, \ldots, z_{i,K}]^{\mathsf{T}}$ *is the output of GNNs (i.e., the prediction probability), and* $\hat{y}_i = \arg\max_k z_{i,k}$ *and* $\hat{p}_i = \max_k z_{i,k}$ *are the prediction and the confidence respectively. Then we define* $f_\theta$ *to be perfectly calibrated as:*

$$\mathbb{P}(\hat{y}_i = y_i | \hat{p}_i = p) = p, \forall p \in [0, 1]. \tag{1}$$

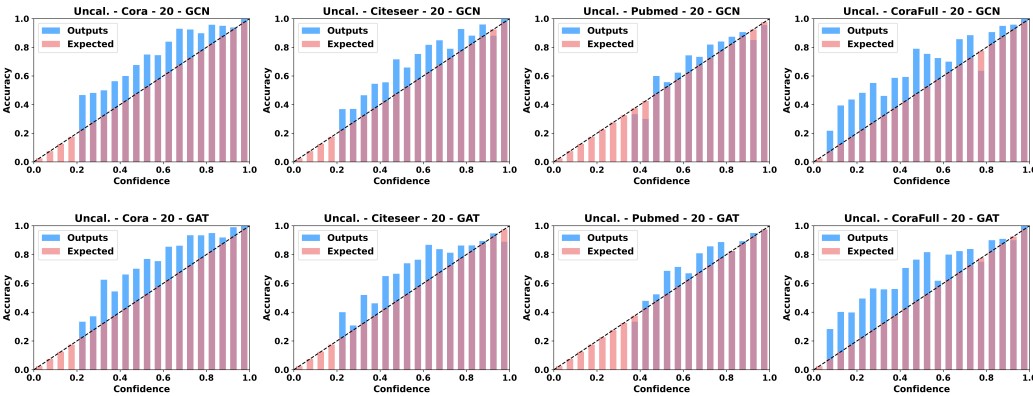

Figure 1: Reliability diagrams for GCN (top) and GAT (bottom) without confidence calibration. The diagram is expected to plot an identity function of accuracy with respect to confidence. Any deviation from a perfectly diagonal (i.e., the difference between blue and red histogram) represents the miscalibration.

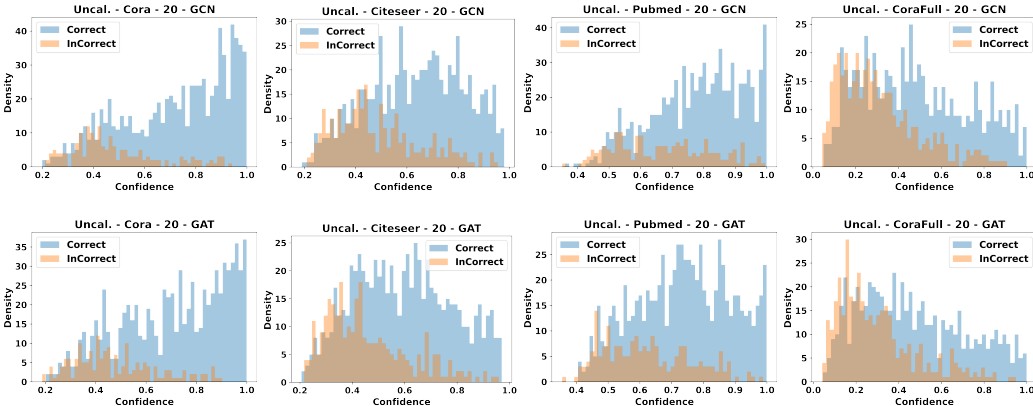

Figure 2: Confidence distribution before calibration.

According to Definition 1, GNN is perfectly calibrated only when the confidence $\hat{p}_i$ is exactly equal to the true probability of getting a correct prediction for every node.

Next, we take two representative GNNs (GCN [16] and GAT [33]) as examples to analyze whether they are perfectly calibrated. Specifically, we apply GCN and GAT to four widely used datasets Cora [29], Citeseer [29], Pubmed [29], CoraFull [3], and examine whether their results satisfy Definition 1. To provide more results, we select three label rates for training set (i.e., 20, 40, 60 labeled nodes per class). All the experimental settings follow [16, 33]. Since the true probability $p$ cannot be exactly known, we take an approximate way to evaluate the calibration as in [12]. In particular, we first partition the [0,1] range of confidence into 20 equal bins and then we group the nodes into corresponding bins according to their confidence. After that we calculate the average accuracy of each bin. We expect the average accuracy is equal to the average confidence of each bin, which means the model is approximately perfectly calibrated. For example, if the average confidence of nodes in the bin [0.95, 1.0] is 0.96, and then the classification accuracy in this bin should be 96%.

We illustrate the results of label rate being 20 in Fig. 1 using Reliability Diagrams [23] here, where the x-axis is the confidence in 20 bins of equal size and y-axis is the average accuracy in each bin. The blue represents the classification accuracy of GCN and GAT while the red is our expectation. More results of label rate being 40, 60 and other GNN models can be seen in Fig. 8, Fig. 9, Fig. 12, Fig. 13 and Fig. 14 in the appendix. We can see that in all the datasets, the average accuracy of most bins is higher than the average confidence. In other words, these GNNs actually achieve remarkable

performance, but they all output low confidence, i.e., the GNNs are usually under-confident. Please note that this phenomenon of GNNs is very different from other modern neural networks, which are generally known to be over-confident [12, 19]. Moreover, as shown in Fig. 2, we also visualize the confidence distribution of test nodes, where the x-axis is the confidence and y-axis is the density [28]. The histogram height multiplied by the width is equal to the frequency. The blue represents the confidence distribution of correct predictions while the yellow represents that of incorrect predictions. More results of label rate being 40 and 60 can be seen in Fig. 10 and Fig. 11 in the appendix. We can see that a large quantity of correct predictions are distributed in the low confidence range. The results above indicate that the current GNNs are far from perfect calibration, leading to unreliable confidence.

## 3    Confidence Calibration on GCNs

In this section, we propose our method to calibrate current GNNs. Given $\mathbf{A}$ and $\mathbf{X}$, for a $l$-layer GCN [16], the output of the GCN before the softmax layer can be obtained by:

$$\mathbf{V} = \mathbf{A}\sigma(\cdots \mathbf{A}\sigma(\mathbf{A}\mathbf{X}\mathbf{W}^{(1)})\mathbf{W}^{(2)}\cdots)\mathbf{W}^{(l)} = [\mathbf{v}_1, \cdots, \mathbf{v}_N]^\mathsf{T}, \quad (2)$$

where $\mathbf{W}^{(l)}$ is the weight matrix of $l$-th layer in GCN and $\sigma(\cdot)$ is the activation function. For each node $i \in \{1, \cdots, N\}$, our goal is to learn a calibration function which is fed with $\mathbf{v}_i$ (often known as the logit of node $i$) and outputs a calibrated confidence using a held-out dataset in a post-hoc way. The calibration function should satisfy three points below: (1) taking the network topology into account (2) non-linear (3) preserving the classification accuracy of the GCNs.

### 3.1    CaGCN: GCNs as Calibration Function

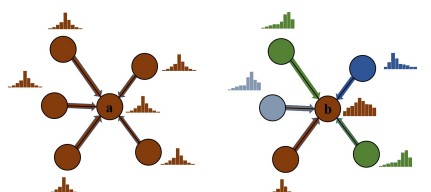

Figure 3: The illustration of the confidence propagation. Different colors indicate different classes.

Table 1: Summary of total variation of confidence before and after calibration (Bold: best). Uncal. is short for uncalibrated and TS is short for temperature scaling.

| Dataset | GCN | | |
|---|---|---|---|
| | Uncal. | TS | Ours |
| Cora | 240.267 | 172.346 | **164.651** |
| Citeseer | 128.145 | 112.212 | **108.684** |
| Pubmed | 1299.33 | 1266.68 | **1113.41** |
| CoraFull | 6014.32 | 4698.20 | **4500.30** |

We assume that the ground-truth confidence distribution in a graph has homophily property, i.e., the confidence of neighboured nodes given by well-calibrated models should be similar, and thus we conduct an experiment to verify this. We employ the classic temperature scaling method [12] as our calibration function and use the total variation [27] of confidence as our evaluation, which sums the difference of confidence between all the neighboured nodes. We compare the total variation of confidence before and after confidence calibration, where the results are shown in Table 1. We can find that the total variation of confidence does decrease after temperature scaling, which verifies our assumption. This inspires us that if a GCN model is well-calibrated, then the confidence between neighbors should be more similar than before.

To this end, we find that GCN itself can play the role of calibration function that meets above requirement since GCN is able to propagate node features along the network topology and smooth similar information between neighboured nodes. Therefore, we can employ another $l$-layer GCN (CaGCN) as our calibration function to propagate the confidence along the network topology. Specifically, given the output $\mathbf{V}$ of the classification GCN, the logit $\mathbf{v}'_i$ and confidence $\hat{p}_i$ for node $i$ after calibration can be obtained by:

$$\begin{aligned} \mathbf{V}' &= \mathbf{A}\sigma(\cdots \mathbf{A}\sigma(\mathbf{A}\mathbf{V}\mathbf{W}^{(1)})\mathbf{W}^{(2)}\cdots)\mathbf{W}^{(l)} = [\mathbf{v}'_1, \cdots, \mathbf{v}'_N]^\mathsf{T}, \\ \mathbf{z}_i &= [\sigma_{SM}(v'_{i,1}), \cdots, \sigma_{SM}(v'_{i,K})]^\mathsf{T}, \hat{p}_i = \max_k z_{i,k}, \end{aligned} \quad (3)$$

where $\sigma_{SM}(v'_{i,\cdot}) = \frac{\exp(v'_{i,\cdot})}{\sum_{j=1}^{K} \exp(v'_{i,j})}$ is the softmax operation. Then the total variation of confidence will surely become lower and the original classification GCN will be calibrated. Please note that although

temperature scaling can be directly applied here, compared with GCN, it does not take the network topology into account, which may cause mistakes mentioned in Section 1. Moreover, temperature scaling only employs a linear transformation, and GCN is able to learn a non-linear calibration function.

For a comprehensive understanding of confidence propagation, we make a detailed and visible illustration here. As shown in Fig. 3, the logits of two nodes $a$ and $b$ are the same, but node $a$ is similar to its neighbors while node $b$ is not. Apparently, the predictions of GCNs for $a$ should be more confident than $b$. Suppose that $a$, $b$ and their neighbours are under-confident based on the observation above. If we continue to propagate their logits along the topology using another GCN, the logits of $a$ and its neighbors will tend to be the same. Therefore, if one or more of these nodes are calibrated during the calibration process, all of them will be calibrated as well. The confidence is propagated in this way. On the other hand, looking at another node $b$, it is as difficult even for manual classification as it is for GCNs. Consequently, the confidence of $b$ should stay still even be lower. However, it will become higher because of the influence from $a$ if we use the traditional calibration method without considering the network topology. Instead, when the network topology is taken into account, the logit of $b$ will be averaged by its neighboured and each dimension tends to $1/K$. It will be correctly calibrated when other nodes in the same situation are well-calibrated.

## 3.2 The Accuracy-Preserving Property

Until now, we have proposed a non-linear calibration model CaGCN which can take the network topology into account, but the accuracy-preserving property cannot be satisfied. To address this problem, we firstly study the general accuracy-preserving calibration function.

**Proposition 1.** *Let* $h : \mathbb{R}^K \to \mathbb{R}^K$ *be a calibration function,* $s : \mathbb{R} \to \mathbb{R}$ *be a 1-D function and* $\mathbf{v}_i = [v_{i,1}, \cdots, v_{i,K}]^\mathsf{T}$ *be the logit of node $i$. The calibration function $h$ preserves the classification accuracy of the original model if $s$ is a strictly isotonic function and $h$ satisfies:*

$$h(\mathbf{v}_i) = [s(v_{i,1}), \ldots, s(v_{i,K})]^\mathsf{T}, \forall i \in \{1, \cdots, N\}. \tag{4}$$

*Proof* We set $v_{i,1} < v_{i,2} < \cdots < v_{i,K}$ without loss of generality. Since $[s(v_{i,1}), \ldots, s(v_{i,K})]^\mathsf{T}$ shares the same order with $\mathbf{v}_i$ as a result of the strictly isotonicity of $s$, the order between classes of the logit $\mathbf{v}_i$ is unchanged, hence the accuracy of the prediction is preserved. ∎

Temperature scaling [12] is the simplest accuracy-preserving calibration method using a scalar parameter $t$ called *temperature* for all classes. Given the logit $\mathbf{v}_i$ of node $i$, the confidence of the prediction is $\hat{p}_i = \max_k \sigma_{SM}(v_{i,k}/t)(t > 0)$. In temperature scaling, $h(\mathbf{v}_i) = [v_{i,1}/t, \cdots, v_{i,K}/t]^\mathsf{T}$ is the calibration function and $s(x) = x/t$ is the strictly isotonic function.

However, we can find that temperature scaling (TS) [40] only performs the same linear transformation for all the nodes using the same $t$. As mentioned in Eq. 3, we propose to use CaGCN as our calibration function, while CaGCN is generally not isotonic, i.e., the order between classes of $\mathbf{v}_i$ and $\mathbf{v}'_i$ is not the same, implying that after calibration by CaGCN, the accuracy of original GCN cannot be preserved. Instead, here we propose an improved CaGCN. Given the output $\mathbf{V}$ of the classification GCN, we firstly use a $l$-layer GCN to learn a unique temperature $t_i$ for each node $i$, then get a calibrated logit $\mathbf{v}'_i$ by transforming its original logit $\mathbf{v}_i$ using $t_i$ in a temperature-scaling way, and finally obtain calibrated confidence $\hat{p}_i$ as follows:

$$\begin{aligned} \mathbf{t} &= \sigma^+(\mathbf{A}\sigma(\cdots\mathbf{A}\sigma(\mathbf{AVW}^{(1)})\mathbf{W}^{(2)}\cdots)\mathbf{W}^{(l)}) = [t_1, \cdots, t_N]^\mathsf{T} (t_i > 0, \forall i \in \{1, \cdots, N\}), \\ \mathbf{v}'_i &= h(\mathbf{v}_i, t_i) = [v_{i,1}/t_i, \cdots, v_{i,K}/t_i]^\mathsf{T}, \mathbf{z}_i = [\sigma_{SM}(v'_{i,1}), \cdots, \sigma_{SM}(v'_{i,K})]^\mathsf{T}, \hat{p}_i = \max_k z_{i,k}, \end{aligned} \tag{5}$$

where $t_i \in \mathbb{R}$ is a scalar greater than zero and $\sigma^+(\mathbf{x}) = log(1 + exp(\mathbf{x}))$ is an element-wise softplus activation [8]. The model proposed in Eq. 5 does not change the order between classes of $\mathbf{v}_i$ and $\mathbf{v}'_i$, implying that the accuracy of original GCN is preserved. Compared Eq. 5 with Eq. 3, we can find that Eq. 5 makes the same transformation on all the dimensions of $\mathbf{v}_i$, which will limit the learnable calibration function space. However, we will prove that actually Eq. 5 is the same with the model proposed in Eq. 3 on confidence calibration using the Proposition 2. Considering that for any logit $\mathbf{v}_i$, our expectation is in fact that the calibration model can output any confidence $\hat{p}_i \in (\frac{1}{K}, 1)$. Please note that $\hat{p}_i \geq \frac{1}{K}$, or the prediction will be changed. Since Eq. 3 has no limitation on the learnt calibration model, its output $\hat{p}_i$ can take any value from $\frac{1}{K}$ to 1. Therefore, if we can prove the output $\hat{p}_i$ in Eq. 5 can also traverse the interval $(\frac{1}{K}, 1)$ for any $\mathbf{v}_i$, the equality between Eq. 3 and Eq.5 can be proved.

**Proposition 2.** *Given the original logit $\mathbf{v}_i = [v_{i,1}, \cdots, v_{i,K}]^\top$ of node i, assume $v_{i,j}$ not approaching infinity for each $j \in \{1, \cdots, K\}$. The calibrated confidence $\hat{p}_i$ in Eq. 5 can traverse the interval $(\frac{1}{K}, 1)$ for node i.*

*Proof* We set $v_{i,1} > v_{i,2} > \cdots > v_{i,K}$ without loss of generality. For any $\mathbf{v}_i \in \mathbb{R}^K$, with the assumption of $\mathbf{v}_i$ not approaching infinity, we have that

$$\lim_{t \to 0} \hat{p}_i = \lim_{t \to 0} \frac{exp(v_{i,1}/t_i)}{\sum_{j=1}^K exp(v_{i,j}/t_i)} = \lim_{t \to 0} \frac{exp((v_{i,1} - v_{i,2})/t_i)}{exp((v_{i,1} - v_{i,2})/t_i) + \sum_{j=2}^K exp((v_{i,j} - v_{i,2})/t_i)} = 1 \quad (6)$$

and

$$\lim_{t \to +\infty} \hat{p}_i = \lim_{t \to +\infty} \frac{exp(v_{i,1}/t_i)}{\sum_{j=1}^K exp(v_{i,j}/t_i)} = \frac{1}{K}. \quad (7)$$

Obviously, both $\sigma_{SM}(v_{i,k})$ and $\mathbf{v}_i/t_i$ are continuous, thus $\sigma_{SM}(v_{i,k}/t_i)$ is continuous. Therefore, $\hat{p}_i = \max_k z_{i,k} = \max_k \sigma_{SM}(v_{i,k}/t_i)$ can traverse the interval $(1/K, 1)$. ∎

The assumption about $\mathbf{v}_i$ is easy to be satisfied since the L2-norm in GCN drives the weight matrix $\mathbf{W}$ approaching zero matrix and each element in node feature matrix $\mathbf{X}$ is not infinity. Therefore, based on Eq. 2, each element $v_{i,j}$ in $\mathbf{V}$ cannot approach infinity. From Proposition 2 we know that for any $\mathbf{v}_i$, there exactly exists such a unique temperature $t_i$ that $\hat{p}_i$ can take any value from $1/K$ to 1. In other words, the model can be perfectly calibrated.

### 3.3 Optimization Objective

Since NLL loss [10] can be decomposed into calibration loss and refinement loss [21], minimizing NLL loss benefits for confidence calibration. Therefore, we employ the NLL loss as our objective function with an additional regularization term. We use the prediction probability $\mathbf{z}_i \in \mathbb{R}^K$ in Eq. 5 to calculate the NLL loss. Denote the $K$-class one-hot label for node $i$ as $\mathbf{y}_i = [y_{i,1}, \cdots, y_{i,K}]^\top$ and suppose the size of the validation set is $|D_{val}|$. Then the NLL loss over all validation nodes is represented as $\mathcal{L}_{nll}$ where:

$$\mathcal{L}_{nll} = -\sum_{i=1}^{|D_{val}|} \sum_{k=1}^K y_{i,k} log(z_{i,k}). \quad (8)$$

Due to the under-confidence of GCNs, our goal is to increase the confidence of correct predictions while decreasing that of incorrect predictions. Considering that for incorrect predictions, the NLL loss cannot directly reduce their confidence, therefore, we design a regularization term for NLL loss as follows:

$$\mathcal{L}_{cal} = \frac{1}{n}\left(\sum_{i=1}^{|cor|} 1 - z_{i,m}^{(cor)} + z_{i,s}^{(cor)} + \sum_{i=1}^{|inc|} z_{i,m}^{(inc)} - z_{i,s}^{(inc)}\right), \quad (9)$$

where $|cor|$ and $|inc|$ are the number of nodes correctly and incorrectly predicted and $z_{i,m}$ and $z_{i,s}$ are the max and submax prediction probability. Intuitively, the confidence of incorrect predictions is decreased by reducing the gap between the max and the submax value of $\mathbf{z}_i$ and vice versa. Combining $\mathcal{L}_{nll}$ and $\mathcal{L}_{cal}$, we have the following overall objective function:

$$\mathcal{L} = \mathcal{L}_{nll} + \lambda \mathcal{L}_{cal}, \quad (10)$$

where $\lambda$ is the parameter of the regularization term. With the guide of labeled data, we can optimize CaGCN via back propagation and learn the calibrated confidence. The overall framework of CaGCN is shown in Fig. 4.

## 4 Self-training with Confidence Calibration

Here we propose a practical application of confidence calibration to improve the performance of self-training in GCNs. Self-training is to predict the labels for unlabeled data, and then add them to the training set, so as to achieve better performance. When applying self-training to GCN, we firstly obtain the predictions $\hat{y}_i$ and the confidence $\hat{p}_i$ given by GCN and then add the most confident nodes to the training set with pseudo labels $\hat{y}_i$ based on $\hat{p}_i$. We continue to train until convergence. However, existing self-training methods perform not as expected with higher label rates [30]. Considering the under-confidence of existing GCNs, motivated by [26], we argue that the under-performance of

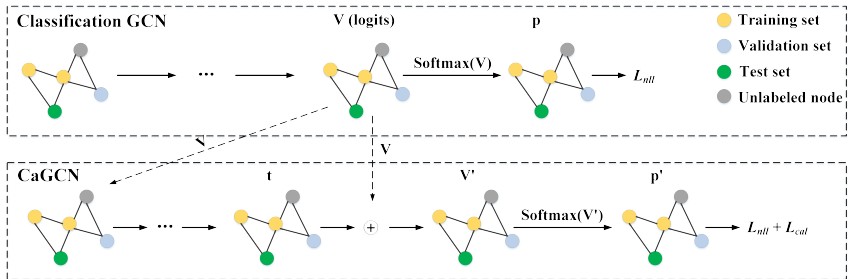

Figure 4: The overall framework of CaGCN. Solid lines represent that we can backpropagate gradient here while dashed lines represent we cannot. We firstly train a classification GCN using the training set to obtain the logit **V** of all the nodes. Then we feed **V** to CaGCN to get the temperature **t** and transform **V** using **t** into **V′**. Finally, the loss can be obtained using **V′** after softmax according to Eq. 10 and CaGCN can be optimized with the guide of the validation set.

Table 2: ECE (M=20) on different models and citation networks of various label rate (L/C) with and without calibration. Uncal. represents the uncalibrated model, (-) denotes this method cannot converge to a meaningful result and bold denotes the best result, the subscript of each result refers to the standard deviation ($\times 10^{-3}$) while the superscript refers to the results of paired t-test ( * for 0.05 level and ** for 0.01 level).

| Dataset | L/C | GCN | | | | GAT | | | |
|---|---|---|---|---|---|---|---|---|---|
| | | Uncal. | TS | MS | CaGCN | Uncal. | TS | MS | CaGCN |
| Cora | 20 | $0.1347_{6.3}$ | $0.0488_{5.5}$ | $0.0414_{5.7}$ | $\mathbf{0.0401}_{6.7}$ | $0.1558_{8.9}$ | $0.0717_{9.8}$ | $0.0544_{9.4}$ | $\mathbf{0.0450}^{**}_{5.6}$ |
| | 40 | $0.1134_{4.7}$ | $0.0417_{7.2}$ | $0.0372_{4.6}$ | $0.0407_{5.4}$ | $0.1340_{5.4}$ | $0.0485_{7.7}$ | $0.0491_{6.0}$ | $\mathbf{0.0365}^{**}_{5.6}$ |
| | 60 | $0.0937_{4.9}$ | $\mathbf{0.0355}_{5.4}$ | $0.0364_{6.1}$ | $0.0376_{4.4}$ | $0.1201_{3.3}$ | $0.0393_{6.1}$ | $0.0411_{5.3}$ | $\mathbf{0.0313}^{**}_{3.2}$ |
| Citeseer | 20 | $0.1248_{7.1}$ | $0.0641_{8.7}$ | $0.0644_{3.7}$ | $\mathbf{0.0595}^{*}_{7.2}$ | $0.1534_{5.0}$ | $0.0916_{8.7}$ | $0.0633_{9.8}$ | $\mathbf{0.0572}_{6.8}$ |
| | 40 | $0.0957_{7.7}$ | $0.0601_{4.2}$ | $\mathbf{0.0538}_{5.7}$ | $0.0545_{5.5}$ | $0.1252_{8.7}$ | $0.0797_{3.1}$ | $0.0590_{5.4}$ | $\mathbf{0.0532}^{*}_{5.4}$ |
| | 60 | $0.0806_{6.4}$ | $0.0559_{5.0}$ | $\mathbf{0.0521}_{6.4}$ | $0.0546_{3.4}$ | $0.1090_{5.9}$ | $0.0648_{7.1}$ | $0.0519_{9.1}$ | $\mathbf{0.0525}_{7.6}$ |
| Pubmed | 20 | $0.0586_{7.7}$ | $0.0541_{3.8}$ | $0.0476_{4.2}$ | $\mathbf{0.0405}^{*}_{6.0}$ | $0.0835_{3.1}$ | $0.0656_{4.6}$ | $0.0501_{3.7}$ | $\mathbf{0.0356}^{**}_{6.3}$ |
| | 40 | $0.0444_{5.5}$ | $0.0446_{6.3}$ | $0.0436_{6.3}$ | $\mathbf{0.0402}^{*}_{4.0}$ | $0.0869_{4.6}$ | $0.0658_{6.5}$ | $0.0539_{6.0}$ | $\mathbf{0.0308}^{**}_{5.4}$ |
| | 60 | $0.0445_{9.7}$ | $0.0367_{6.0}$ | $0.0318_{6.4}$ | $\mathbf{0.0311}_{4.8}$ | $0.0993_{4.1}$ | $0.0669_{6.3}$ | $0.0483_{5.7}$ | $\mathbf{0.0308}^{**}_{5.2}$ |
| CoraFull | 20 | $0.1986_{6.1}$ | $0.1013_{6.1}$ | - | $\mathbf{0.0776}^{**}_{6.4}$ | $0.2119_{3.6}$ | $0.1101_{5.1}$ | - | $\mathbf{0.0788}^{**}_{6.0}$ |
| | 40 | $0.2321_{5.4}$ | $0.1117_{6.5}$ | - | $\mathbf{0.0701}^{**}_{3.9}$ | $0.2438_{4.2}$ | $0.1133_{8.3}$ | - | $\mathbf{0.0738}^{**}_{4.8}$ |
| | 60 | $0.2337_{4.0}$ | $0.0981_{3.8}$ | - | $\mathbf{0.0768}^{**}_{3.4}$ | $0.2497_{1.8}$ | $0.1133_{5.2}$ | - | $\mathbf{0.0849}^{**}_{6.9}$ |

existing self-training methods originals from large numbers of high-accuracy predictions distributing in low-confidence intervals as shown in Fig. 2, causing that they cannot be added to the training set.

Consequently, we design a self-training model CaGCN-st where confidence is firstly calibrated then employed to generate pseudo labels for unlabeled nodes. Specifically, given an unlabeled dataset $D_U$ and a labeled dataset $D_L$ which has been divided into three parts $D_{train}$, $D_{val}$ and $D_{test}$, we firstly train a classification GCN using $D_{train}$ to get the logit of each node. Then all the logits will be fed into a CaGCN to train and we get a calibrated confidence for each node. It should be noted that instead of $D_{val}$, we still employ $D_{train}$ to train our CaGCN. After that, the most confident predictions of $D_U$ will be adopted as the pseudo labels according to a threshold $\tau$ and added to the label set. The $D_{train}$ is enlarged in this way. The process above will be repeated $s$ stages until convergence. Please note that our classification GCN and CaGCN are re-initialized in each stage.

## 5 Experiments

In this section, we evaluate the performance of CaGCN on confidence calibration and CaGCN-st on self-training respectively. We choose the commonly used *citation networks* Cora [29], Citeseer [29], Pubmed [29] and CoraFull [3] for evaluation, and more detailed descriptions are in Appendix B.

Table 3: Node classification accuracy and the standard deviation on GCN and its self-training variants.

| Dataset | L/C | Methods | | | | | | |
|---|---|---|---|---|---|---|---|---|
| | | Orig. | St. | Ct. | Union | Inter. | TS-st | CaGCN-st |
| Cora | 20 | $81.63_{0.24}$ | $82.27_{0.33}$ | $81.51_{0.30}$ | $81.85_{0.68}$ | $81.41_{0.28}$ | $82.68_{0.20}$ | $\mathbf{83.11}^{*}_{0.52}$ |
| | 40 | $83.99_{0.26}$ | $83.59_{0.34}$ | $83.66_{0.25}$ | $83.33_{0.41}$ | $83.38_{0.33}$ | $\mathbf{84.44}_{0.35}$ | $84.37_{0.38}$ |
| | 60 | $84.44_{0.29}$ | $84.98_{0.32}$ | $84.63_{0.31}$ | $85.03_{0.30}$ | $84.88_{0.18}$ | $85.60_{0.24}$ | $\mathbf{85.79}_{0.27}$ |
| Citeseer | 20 | $71.64_{0.32}$ | $73.24_{0.44}$ | $74.22_{0.29}$ | $74.60_{0.38}$ | $72.25_{0.45}$ | $74.20_{0.24}$ | $\mathbf{74.90}^{**}_{0.40}$ |
| | 40 | $72.25_{0.32}$ | $74.70_{0.33}$ | $72.12_{0.39}$ | $74.79_{0.36}$ | $73.66_{0.32}$ | $\mathbf{75.62}_{0.19}$ | $75.48_{0.50}$ |
| | 60 | $73.20_{0.35}$ | $75.08_{0.29}$ | $73.21_{0.36}$ | $75.53_{0.30}$ | $75.23_{0.23}$ | $75.87_{0.24}$ | $\mathbf{76.43}^{**}_{0.20}$ |
| Pubmed | 20 | $79.57_{0.33}$ | $80.32_{0.18}$ | $79.67_{0.32}$ | $81.12_{0.29}$ | $79.59_{0.29}$ | $80.95_{0.18}$ | $\mathbf{81.16}^{*}_{0.36}$ |
| | 40 | $80.65_{0.39}$ | $82.20_{0.32}$ | $81.62_{0.40}$ | $81.84_{0.23}$ | $80.46_{0.55}$ | $82.28_{0.39}$ | $\mathbf{83.08}^{*}_{0.21}$ |
| | 60 | $83.38_{0.34}$ | $83.35_{0.28}$ | $83.40_{0.36}$ | $83.32_{0.35}$ | $83.31_{0.17}$ | $83.26_{0.39}$ | $\mathbf{84.47}^{**}_{0.23}$ |
| CoraFull | 20 | $60.45_{0.43}$ | $60.87_{0.28}$ | $60.12_{0.45}$ | $60.52_{0.35}$ | $61.01_{0.53}$ | $61.73_{0.41}$ | $\mathbf{62.19}^{*}_{0.49}$ |
| | 40 | $65.77_{0.37}$ | $65.83_{0.45}$ | $64.22_{0.35}$ | $64.33_{0.42}$ | $65.84_{0.37}$ | $66.11_{0.60}$ | $\mathbf{66.30}^{*}_{0.31}$ |
| | 60 | $66.52_{0.25}$ | $66.62_{0.30}$ | $66.64_{0.29}$ | $66.78_{0.29}$ | $66.82_{0.32}$ | $66.95_{0.45}$ | $\mathbf{67.60}^{*}_{0.40}$ |

## 5.1 Confidence Calibration Evaluation

**Baselines.** Since our CaGCN is a general calibration model for GNNs, here we choose GCN [16] and GAT [33] as our classification models. For comparison, we choose the classic post-hoc calibration methods temperature scaling (TS) [12] and matrix scaling with off-diagonal regularization (MS) [18] as our baselines.

**Experimental settings.** For the base model GCN and GAT, i.e., the uncalibrated model, we follow parameters suggested by [16] and [33] and further carefully tune them to get optimal performance. For the post-hoc calibration technique, we follow the official implementation [12, 18]. For our CaGCN, we train a two-layer GCN with the hidden layer dimension to be 16. We set $\lambda = 0.5$ for all datasets, weight decay to be 5e-3 for Cora, Citeseer, Pubmed and 0.03 for CoraFull. Other parameters of CaGCN follows [16]. We evaluate the performance of confidence calibration by ECE [22], NLL [10] and Brier Score (BS) [4], which we expect are smaller, and we set the bin number $M = 20$ for ECE (more details can be seen in Appendix A). For all methods, we randomly run 10 times and report the average results. More detailed experimental settings can be seen in Appendix B.

**Results.** Table 2 reports calibration results evaluated by ECE (more results on NLL and Brier Score are in Appendix C.1). We have the following observations: (1) Compared with uncalibrated models and other baselines, CaGCN is statistically significantly better at the * 0.05 level and ** 0.01 level. (2) The ECE values on uncalibrated models are generally the highest, implying that GCN and GAT are poorly calibrated. (3) MS behaves badly on datasets with many classes, e.g., CoraFull. This is because the number of parameters for matrix scaling scales quadratically with the number of classes while the size of the validation set keeps unchanged. Therefore, it will over-fit to the small validation set when dataset has a great number of classes. However, CaGCN does not have this problem.

**Additional analysis.** In Section 2 we visualize the under-confidence problem of existing GNNs using reliability diagrams. Here we utilize the same visualization method to make a comparison before and after confidence calibration. As shown in Fig. 8, Fig. 9, Fig. 10 and Fig. 11 in the appendix, we can find that the confidence is well-calibrated after calibration.

## 5.2 Classification Evaluation of Self-Training

**Baselines.** Since self-training can be applied to any models, here we choose GCN and GAT as our base models, i.e., the original models (Orig.) without self-training, and we choose self-training (St.), co-training (Ct.), Union, Intersection (Inter.) methods proposed in [20] for comparison, which are commonly used as the baselines in self-training. Furthermore, we employ TS as the confidence calibration function in CaGCN-st as another baseline and we denote it by TS-st.

**Experimental settings.** We set the learning rate $lr = 0.001$ for CaGCN-st and train our CaGCN-st 200 epochs for Cora, 150 epochs for Citeseer, 100 epochs for Pubmed and 500 epochs for CoraFull. We set the threshold $\tau \in \{0.8, 0.85, 0.9, 0.95, 0.99\}$ and the maximum number of stage $s = 10$. As

for baselines, all the parameters follow [20] and we further carefully tune them to get optimal performance. For all methods, we randomly run 10 times and report the average results.

**Results.** Table 3 summarizes the node classification accuracy on GCN and its self-training variants. More results on GAT can be seen in Appendix C.2. We have the following observation: (1) CaGCN-st consistently outperforms all the baselines on all the datasets and label rates at the * 0.05 level. (2) Compared with the base model, self-training methods generally achieve better results, which proves their effectiveness. (3) Self-training methods with confidence calibration (i.e., TS-st and CaGCN-st) have better performance, which implies that confidence calibration scales more correct predictions to the high confidence range while keeps incorrect predictions basically unchanged, which we believe is beneficial for self-training.

**Ablation study.** CaGCN-st generates pseudo labels based on the calibrated confidence. Here we study the effectiveness of the confidence calibration function CaGCN in CaGCN-st. We propose a variant GCN-st of CaGCN-st, where CaGCN is removed from CaGCN-st while other parts are kept unchanged. All the experimental settings of GCN-st are the same as CaGCN-st. We report the results in Table 4, and we can observe that CaGCN-st consistently outperforms GCN-st on all the datasets, implying that self-training with calibrated confidence can generate more correct pseudo labels.

**Additional analysis.** We also investigate the changing trends of accuracy with respect to the threshold $\tau$ in CaGCN-st in Appendix C.2 and study why GCNs are poorly calibrated in Appendix D.

Table 4: Abaltion study on self-training

| Dataset | L/C | GCN | | GAT | |
|---|---|---|---|---|---|
| | | GCN-st | CaGCN-st | GCN-st | CaGCN-st |
| Cora | 20 | 82.28 | **83.11** | 84.08 | **84.08** |
| | 40 | 84.10 | **84.37** | 85.50 | **85.63** |
| | 60 | 85.16 | **85.79** | 85.57 | **86.26** |
| Citeseer | 20 | 74.13 | **74.90** | 73.73 | **74.34** |
| | 40 | 75.28 | **75.48** | 75.07 | **75.62** |
| | 60 | 75.85 | **76.43** | 75.13 | **76.08** |
| Pubmed | 20 | 81.01 | **81.16** | 80.34 | **81.17** |
| | 40 | 82.90 | **83.08** | 82.75 | **83.47** |
| | 60 | 83.44 | **84.47** | 83.46 | **83.95** |
| CoraFull | 20 | 61.32 | **62.19** | 62.09 | **65.46** |
| | 40 | 65.96 | **66.30** | 65.92 | **66.86** |
| | 60 | 66.43 | **67.60** | 66.54 | **67.45** |

## 6  Related Work

**Graph Neural Networks.** Modern GCNs mimics CNNs to learn the local and global structural patterns of graphs through designed convolution and readout functions. [5] generalizes CNNs to graph signal based on the spectrum of graph Laplacian. ChebNet [7] uses Chebyshev polynomials to approximate the $K$-order localized graph filters and GCN [16] further employs the 1-order simplification of the Chebyshev filter. GAT [33] utilizes attention mechanisms to adaptively learn aggregation weights. GraphSAGE [13] uses various ways of pooling for aggregation. [20] introduces self-training to GCNs and [30] proposes a multi-stage self-supervised (M3S) self-training algorithm of GCNs. Both [20] and [30] focus on the few-shot learning and neither has ever explored self-training with higher label rates in GCNs. More works on GNNs can be found in surveys [36, 43], however, to the best of our knowledge, current GNNs have not considered the confidence calibration.

**Confidence Calibration.** Confidence calibration has been studied for a long time in CV and NLP [12, 23, 25, 14, 19, 37, 40, 42]. [12] discovers modern neural networks are poorly calibrated and study factors influencing calibration. Platt scaling [24] is a simple post-hoc calibration method for binary models, which transforms the logit using scalar parameters. Temperature scaling is the simplest multi-class extension of Platt scaling and matrix and vector scaling are another two extensions of platt scaling. [40] proposes Mix-n-Match calibration strategies which mix parameter methods with non-parameter methods. [25] explores the non-linear space for post-hoc calibration function using a neural network. Moreover, [32] points out GNNs can be miscalibrated in the supervised scenario and mainly focus on miscalibration originated from the imbalanced class distribution. However, none of them have considered the confidence calibration in GNNs on the common semi-supervised scenario.

## 7  Conclusion

Current efforts on advancing GNNs mostly focus on classification accuracy. However, when deploying GNNs to real-world applications, especially safety-critical fields, whether the results of GNNs are

trustworthy is another important factor that cannot be neglected. In this paper, we study the confidence calibration problem in GNNs and discover existing GNNs are under-confident on their predictions. To solve this problem, we propose a novel trustworthy GNN model CaGCN which respects the homophily property of confidence in GNNs and preserves the classification accuracy. Moreover, we propose a novel self-training method CaGCN-st where confidence is first calibrated by CaGCN and then used to generate pseudo labels. Extensive experiments demonstrate the effectiveness of our proposed model in terms of both calibration and accuracy.

An interesting direction for future work is to extend CaGCN to other graph tasks, but more studies need to be conducted. We take the link prediction as an example, where we can regard the link prediction as a binary classification problem and the output as the confidence. Considering that the ground-truth confidence distribution for nodes should have the homophily property as is shown in Section 3.1, edges are likely to have the same property as well. As a result, we can employ CaGCN to propagate the confidence between edges by regarding the edges as the nodes. However, more exploration still needs to be conducted for the homophily property of edges.

**Broader impact.** Current efforts on advancing GNNs mostly focus on classification accuracy. However, when deploying GNNs to real-world applications, especially safety-critical fields, whether the results of GNNs are trustworthy is another important factor. The demands for a trustworthy model are universal and extensive such as in the field of disease prediction [31], traffic states prediction [6] and object detection [11] for autonomous driving, where estimating the true probability of getting a correct prediction is necessary. We take the disease prediction [31] as an example, where GNNs are utilized to encode the information of different symptoms, users and diseases. In this scenario, accurately and comprehensively predicting diseases at an early stage will help patients receive prevention treatments in a timely manner. Otherwise, the misdiagnosis and missed diagnosis will endanger the health of patients. Therefore, a trustworthy model is urgently needed. Our CaGCN can make a trustworthy prediction based on its confidence, and as a result, decrease the risk of misdiagnosis and missed diagnosis. We hope our work can provide insights for future improvements in tackling the trustworthiness problem in other saftey-critical fields.

**Limitations.** One potential issue of this work is that it provides a limited explanation to the under-confidence problem. We advocate peer researchers to look into this, making GNNs more reliable in different domains. Other than that, since this work is mostly on the discovery of the confidence calibration problem in GNNs and the theoretical aspect of improving calibration, we do not foresee any direct negative impacts on the society.

## Acknowledgments and Disclosure of Funding

This work is supported in part by the National Natural Science Foundation of China (No. 62172052, No. U20B2045, U1936104, 61772082, 61702296, 62002029).

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
