# A  Evaluation Metrics

## A.1  Expected Calibration Error (ECE)

Expected calibration error (ECE) [22] is a common calibration metric, which measures the difference in expectation between confidence and accuracy:

$$ECE = \mathbb{E}[|\mathbb{P}(\hat{y}_i = y_i | \hat{p}_i = p) - p|], \tag{11}$$

where $\hat{p}_i$ is the confidence for node $i$, $\hat{y}_i$ is the prediction, $y_i$ is the label and $p$ is the true probability that $\hat{y}_i$ is correctly predicted. However, we cannot exactly know the true probability $p$, thus Eq. 11 cannot be computed directly. The approximation to ECE is generally employed as the metric:

$$ECE = \sum_{m=1}^{M} \frac{|B_m|}{N} |acc(B_m) - conf(B_m)|, \tag{12}$$

where $M$ is the number of equally-spaced bins (similar to the reliability diagrams as mentioned in Section 2 that predictions are partitioned into), $|B_m|$ is the number of predictions falling into the $m$-th bin according to their confidence and $acc(B_m) = \frac{1}{|B_m|} \sum_{i \in B_m} \mathbb{1}(y_i = \hat{y}_i)$, $conf(B_m) = \frac{1}{|B_m|} \sum_{i \in B_m} \hat{p}_i$ represent the average accuracy and confidence in each bin respectively. The difference between $acc$ and $conf$ can be intuitively seemed as the deviation of the outputs to the diagonal in Fig. 1.

## A.2  Brier Score (BS)

Brier Score (BS) [4] is another commonly used calibration metric, which measures the accuracy of probabilistic predictions. The higher the accuracy of predictions is, the lower BS is. For any given prediction $\hat{y}_i$, BS is the lowest when the prediction probability $\mathbf{z}_i$ is exactly equal to the true probability that $\hat{y}_i$ is correct. Given the one-hot label $\mathbf{y}_i$ for node $i$, BS can be represented as follows:

$$BS = \frac{1}{N} \sum_{i=1}^{N} \sum_{k=1}^{K} (z_{i,k} - y_{i,k})^2. \tag{13}$$

# B  More Experimental Details

## B.1  Datasets and Environment

We choose the commonly used Cora [29], Citeseer [29], Pubmed [29] and CoraFull [3] for evaluation, where nodes represent papers, edges are the citation relationship between papers, node features are comprised of bag-of-words vector of the papers and labels represent the fields of papers. We choose 500 nodes for validation, 1000 nodes for test and select three label rates for the training set (i.e., 20, 40, 60 labeled nodes per class). The details of these datasets are summarized in Table 5. Our data are public and do not contain personally identifiable information and offensive content. The address of our data is `https://docs.dgl.ai/en/latest/api/python/dgl.data.html#node-prediction-datasets` and the license is Apache License 2.0. The environment where our code runs is shown as follows:

- Operating system: Linux version 3.10.0-693.el7.x86_64
- CPU information: Intel(R) Xeon(R) Silver 4210 CPU @ 2.20GHz
- GPU information: GeForce RTX 3090

## B.2  Additional Experimental Details for Calibration

For temperature scaling, we follow the official implementation in `https://github.com/gpleiss/temperature_scaling` with MIT license. The learning rate is 0.01 and the maximum number of iteration is 50. For matrix scaling, we add an additional off-diagonal regularization term [18] in case of overfitting. The official implementation is in `https://github.com/dirichletcal/experiments_neurips` and we implement it in Pytorch. The learning rate is 0.01 and the maximum number of iteration is 400.

Table 5: The statistics of the datasets.

| Dataset | #Nodes | #Edges | #Classes | #Features | #Training | #Validation | #Test |
|---------|--------|--------|----------|-----------|-----------|-------------|-------|
| Cora | 2708 | 5429 | 7 | 1433 | 140/280/420 | 500 | 1000 |
| Citeseer | 3327 | 4732 | 6 | 3703 | 120/240/360 | 500 | 1000 |
| Pubmed | 19717 | 44338 | 3 | 500 | 60/120/180 | 500 | 1000 |
| CoraFull | 19793 | 65311 | 70 | 8710 | 1400/2800/4200 | 500 | 1000 |

### B.3 Additional Experimental Details for Self-Training

For the four baselines self-training, co-training, union, intersection proposed in [20], the official implementation is in `https://github.com/Davidham3/deeper_insights_into_GCNs` and we implement them in Pytorch. For our new proposed TS-st, the learning rate is 0.001, maximum number of iteration is 50 for Cora, Citeseer, Pubmed and 25 for CoraFull. More detailed experimental settings can be seen in Table 9.

### B.4 Other Source Code

The acquisition of all the code below complies with the provider's license and do not contain personally identifiable information and offensive content. The address of code of baselines are listed as follows:

GCN (MIT license): `https://github.com/tkipf/pygcn`

GAT (MIT license): `https://github.com/Diego999/pyGAT`

Our code can be found in the supplemental material and all the related experimental details (e.g., environment, experimental settings for our methods and all the baselines) are included in README.

## C  Additional Results

### C.1  Additional Results for Calibration

**NLL and BS.** Since NLL and BS can be affected by the accuracy of predictions, we should keep the accuracy unchanged when evaluating the performance of calibration methods. However, the accuracy-preserving property cannot be satisfied by MS. Therefore, we omit it from our baselines for fairness. Table 6 and Table 7 report calibration results evaluated by NLL and BS. We find that CaGCN is generally better than other baselines at * 0.05 level and ** 0.01 level.

**Reliability diagrams.** In Section 2 we visualize the under-confidence problem of existing GNNs using reliability diagram. Here we still employ the same method to make a comparison before and after confidence calibration. Fig. 8 and Fig. 9 show the reliability diagrams on different models and networks of various label rates before (odd rows) and after (even rows) calibration. We can see that the confidence of predictions on all the datasets and models is well-calibrated. Moreover, for further verifying our conclusion that GNNs are under-confident, we also demonstrate the reliability diagrams of another four representative GNNs (GraphSAGE [13], APPNP [17], SGC [35], GIN [38]) on all the datasets with 20, 40, 60 label rates. The results are summarized in Fig. 12, 13, 14. Similarly, we can observe that in almost all the datasets, the average accuracy of most bins is higher than the average confidence, which means these models are also under-confident, verifying our conclusion again.

**Confidence distribution.** In Section 2 we visualize the confidence distribution as a supplement for the under-confidence problem of existing GNNs. Same as before, here we visualize the confidence distribution before (odd rows) and after (even rows) confidence on different models and networks of various label rates to make a comparison. As shown in Fig. 10 and Fig. 11, we can see that a large quantity of correct predictions have been transformed into a higher confidence range while incorrect predictions change little.

Table 6: NLL and the standard deviation ($\times 10^{-3}$) on different models and citation networks of various label rate (L/C).

| Dataset | L/C | GCN | | | GAT | | |
|---|---|---|---|---|---|---|---|
| | | Uncal. | TS | CaGCN | Uncal. | TS | CaGCN |
| Cora | 20 | $0.6680_{4.3}$ | $0.5998_{3.6}$ | $\mathbf{0.5997}_{3.6}$ | $0.6773_{7.1}$ | $0.6014_{6.8}$ | $\mathbf{0.5831}^{**}_{4.9}$ |
| | 40 | $0.5885_{4.4}$ | $0.5356_{4.5}$ | $\mathbf{0.5340}_{4.3}$ | $0.5996_{3.8}$ | $0.5229_{6.2}$ | $\mathbf{0.5155}^{*}_{4.2}$ |
| | 60 | $0.5313_{1.8}$ | $0.4882_{3.0}$ | $\mathbf{0.4832}^{**}_{1.8}$ | $0.5357_{2.8}$ | $0.4667_{3.2}$ | $\mathbf{0.4579}^{**}_{2.0}$ |
| Citeseer | 20 | $0.9105_{4.7}$ | $0.8700_{4.6}$ | $\mathbf{0.8683}^{*}_{3.0}$ | $0.9638_{2.3}$ | $0.9157_{5.7}$ | $\mathbf{0.8872}^{**}_{4.0}$ |
| | 40 | $0.8634_{4.5}$ | $0.8385_{3.7}$ | $\mathbf{0.8324}^{**}_{4.2}$ | $0.9094_{3.8}$ | $0.8735_{5.9}$ | $\mathbf{0.8472}^{**}_{5.4}$ |
| | 60 | $0.8203_{2.3}$ | $0.8031_{2.6}$ | $\mathbf{0.7978}^{**}_{4.0}$ | $0.8601_{3.3}$ | $0.8275_{5.2}$ | $\mathbf{0.8137}^{*}_{6.4}$ |
| Pubmed | 20 | $0.5511_{3.2}$ | $0.5479_{2.6}$ | $\mathbf{0.5454}_{2.7}$ | $0.5748_{2.2}$ | $0.5609_{2.4}$ | $\mathbf{0.5416}^{**}_{2.0}$ |
| | 40 | $0.4970_{2.5}$ | $0.4939_{2.6}$ | $\mathbf{0.4934}_{2.1}$ | $0.5234_{3.1}$ | $0.5088_{4.3}$ | $\mathbf{0.4871}^{**}_{3.1}$ |
| | 60 | $0.4527_{2.5}$ | $0.4484_{1.9}$ | $\mathbf{0.4396}^{**}_{2.2}$ | $0.4893_{3.3}$ | $0.4681_{4.1}$ | $\mathbf{0.4439}^{**}_{2.6}$ |
| CoraFull | 20 | $1.6651_{4.0}$ | $1.5292_{6.0}$ | $\mathbf{1.4974}^{**}_{5.5}$ | $1.6743_{5.8}$ | $1.5211_{6.0}$ | $\mathbf{1.4985}^{**}_{9.0}$ |
| | 40 | $1.5019_{5.4}$ | $1.3376_{3.9}$ | $\mathbf{1.3123}^{**}_{7.2}$ | $1.5253_{3.5}$ | $1.3379_{3.9}$ | $\mathbf{1.3045}^{**}_{6.4}$ |
| | 60 | $1.4570_{2.4}$ | $\mathbf{1.2757}_{4.1}$ | $1.2873_{7.4}$ | $1.4852_{3.1}$ | $\mathbf{1.2843}_{6.3}$ | $1.2964_{9.1}$ |

Table 7: BS and the standard deviation ($\times 10^{-3}$) on different models and citation networks of various label rate (L/C).

| Dataset | L/C | GCN | | | GAT | | |
|---|---|---|---|---|---|---|---|
| | | Uncal. | TS | CaGCN | Uncal. | TS | CaGCN |
| Cora | 20 | $0.3048_{2.1}$ | $0.2780_{2.2}$ | $\mathbf{0.2746}^{**}_{1.6}$ | $0.3078_{2.9}$ | $0.2802_{2.5}$ | $\mathbf{0.2712}^{**}_{1.7}$ |
| | 40 | $0.2699_{2.0}$ | $0.2507_{1.8}$ | $\mathbf{0.2486}^{*}_{1.2}$ | $0.2747_{2.0}$ | $0.2493_{2.0}$ | $\mathbf{0.2446}^{*}_{1.6}$ |
| | 60 | $0.2418_{1.3}$ | $0.2264_{1.5}$ | $\mathbf{0.2241}_{0.8}$ | $0.2427_{1.4}$ | $0.2206_{0.8}$ | $\mathbf{0.2179}^{**}_{1.1}$ |
| Citeseer | 20 | $0.4362_{2.7}$ | $0.4191_{3.0}$ | $\mathbf{0.4120}^{**}_{2.6}$ | $0.4602_{1.2}$ | $0.4389_{2.8}$ | $\mathbf{0.4210}^{**}_{2.3}$ |
| | 40 | $0.4097_{2.4}$ | $0.4293_{2.0}$ | $\mathbf{0.4057}^{**}_{2.0}$ | $0.4368_{1.6}$ | $0.4220_{2.5}$ | $\mathbf{0.4111}^{**}_{2.2}$ |
| | 60 | $0.4000_{1.4}$ | $0.3936_{1.4}$ | $\mathbf{0.3915}^{*}_{1.4}$ | $0.4141_{1.4}$ | $0.4019_{2.3}$ | $\mathbf{0.3961}^{*}_{3.1}$ |
| Pubmed | 20 | $0.3130_{2.1}$ | $0.3113_{1.4}$ | $\mathbf{0.3089}^{*}_{1.2}$ | $0.3229_{1.4}$ | $0.3163_{1.8}$ | $\mathbf{0.3070}^{**}_{1.1}$ |
| | 40 | $0.2825_{2.0}$ | $0.2812_{1.9}$ | $\mathbf{0.2797}^{*}_{1.8}$ | $0.2915_{1.9}$ | $0.2846_{2.2}$ | $\mathbf{0.2759}^{**}_{1.7}$ |
| | 60 | $0.2536_{1.5}$ | $0.2514_{1.8}$ | $\mathbf{0.2494}^{*}_{1.3}$ | $0.2671_{1.8}$ | $0.2577_{1.6}$ | $\mathbf{0.2494}^{**}_{1.6}$ |
| CoraFull | 20 | $0.6103_{1.4}$ | $0.5723_{2.0}$ | $\mathbf{0.5601}^{**}_{1.9}$ | $0.6128_{1.3}$ | $0.5690_{1.2}$ | $\mathbf{0.5569}^{**}_{3.0}$ |
| | 40 | $0.5645_{2.6}$ | $0.5135_{1.8}$ | $\mathbf{0.4953}^{**}_{2.1}$ | $0.5720_{0.8}$ | $0.5155_{1.2}$ | $\mathbf{0.4981}^{**}_{1.5}$ |
| | 60 | $0.5527_{0.8}$ | $0.4965_{1.3}$ | $\mathbf{0.4903}^{**}_{1.7}$ | $0.5618_{1.1}$ | $0.4992_{1.8}$ | $\mathbf{0.4907}^{**}_{3.0}$ |

## C.2  Additional Results for Self-Training

**Classification evaluation of self-training on GAT.**  Table 8 reports the node classification accuracy on GAT and its self-training variants. Consistent with the result shown in Table 3, our method still achieves the best results.

**Parameter study.**  We investigate the effect of the threshold $\tau$ in CaGCN-st, i.e., the number of unlabeled nodes added to the training set. Generally speaking, with the increase of $\tau$, fewer but more confident nodes will be chosen. Fig. 5 and Fig. 6 show the changing trends of classification accuracy with respect to $\tau$, where different colors represent different label rates. Basically, both too high and too low threshold will harm the performance, since a higher value will leave out correct predictions while a lower value will introduce many incorrect predictions to the label set. CaGCN-st obtains the best performance when $\tau$ is in the range [0.8, 0.9].

Table 8: Node classification accuracy and the standard deviation on GAT and its self-training variants.

| Dataset | L/C | Method | | | | | | |
|---|---|---|---|---|---|---|---|---|
| | | Orig. | St. | Ct. | Union | Inter. | TS-st | CaGCN-st |
| Cora | 20 | $82.10_{0.25}$ | $83.03_{0.25}$ | $82.16_{0.36}$ | $83.18_{0.41}$ | $81.87_{0.33}$ | $83.62_{0.33}$ | $\mathbf{84.08}^{*}_{0.37}$ |
| | 40 | $83.40_{0.36}$ | $84.90_{0.20}$ | $83.20_{0.26}$ | $83.28_{0.37}$ | $83.76_{0.28}$ | $85.34_{0.26}$ | $\mathbf{85.63}_{0.21}$ |
| | 60 | $84.96_{0.21}$ | $85.60_{0.12}$ | $84.29_{0.27}$ | $84.30_{0.43}$ | $85.10_{0.27}$ | $\mathbf{86.49}_{0.18}$ | $86.26_{0.25}$ |
| Citeseer | 20 | $70.86_{0.41}$ | $73.02_{0.26}$ | $71.58_{0.36}$ | $\mathbf{75.38}_{0.26}$ | $71.44_{0.25}$ | $74.28_{0.29}$ | $74.34_{0.21}$ |
| | 40 | $71.60_{0.21}$ | $74.44_{0.20}$ | $72.26_{0.30}$ | $\mathbf{76.73}_{0.26}$ | $73.00_{0.28}$ | $75.12_{0.22}$ | $75.62_{0.19}$ |
| | 60 | $73.08_{0.19}$ | $75.19_{0.25}$ | $72.63_{0.36}$ | $\mathbf{77.11}_{0.30}$ | $75.36_{0.20}$ | $75.52_{0.29}$ | $76.08_{0.39}$ |
| Pubmed | 20 | $79.35_{0.31}$ | $80.47_{0.28}$ | $79.20_{0.28}$ | $80.25_{0.28}$ | $79.31_{0.29}$ | $80.04_{0.27}$ | $\mathbf{81.17}^{**}_{0.30}$ |
| | 40 | $81.17_{0.30}$ | $82.59_{0.31}$ | $79.11_{0.49}$ | $81.99_{0.31}$ | $81.08_{0.28}$ | $82.29_{0.34}$ | $\mathbf{83.47}^{**}_{0.23}$ |
| | 60 | $83.47_{0.23}$ | $83.87_{0.35}$ | $83.01_{0.19}$ | $82.97_{0.19}$ | $83.12_{0.24}$ | $82.35_{0.15}$ | $\mathbf{83.95}^{**}_{0.47}$ |
| CoraFull | 20 | $60.94_{0.36}$ | $61.19_{0.37}$ | $60.15_{0.53}$ | $61.15_{0.29}$ | $60.81_{0.28}$ | $61.30_{0.37}$ | $\mathbf{65.46}^{**}_{0.54}$ |
| | 40 | $65.46_{0.41}$ | $65.64_{0.56}$ | $65.31_{0.22}$ | $65.63_{0.41}$ | $65.81_{0.59}$ | $65.84_{0.43}$ | $\mathbf{66.86}^{*}_{0.56}$ |
| | 60 | $66.52_{0.30}$ | $66.57_{0.24}$ | $66.46_{0.26}$ | $66.43_{0.49}$ | $66.60_{0.26}$ | $66.29_{0.37}$ | $\mathbf{67.45}^{*}_{0.39}$ |

Table 9: Summary of parameters used in CaGCN-st and TS-st. $\alpha_{cal}$: the parameter for weight decay in CaGCN, $epoch\_st$: the number of epochs for self-training, $s$: the number of stage, $\tau$: threshold.

| Dataset | L/C | GCN | | | | | | | GAT | | | | | | |
|---|---|---|---|---|---|---|---|---|---|---|---|---|---|---|---|
| | | CaGCN-st | | | | TS-st | | | CaGCN-st | | | | TS-st | | |
| | | $\alpha_{cal}$ | $s$ | $\tau$ | $epoch\_st$ | $s$ | $\tau$ | $epoch\_st$ | $\alpha_{cal}$ | $s$ | $\tau$ | $epoch\_st$ | $s$ | $\tau$ | $epoch\_st$ |
| Cora | 20 | 5e-3 | 4 | 0.8 | 200 | 3 | 0.8 | 50 | 5e-3 | 6 | 0.8 | 200 | 3 | 0.8 | 50 |
| | 40 | 5e-3 | 2 | 0.8 | 200 | 6 | 0.8 | 50 | 5e-3 | 4 | 0.9 | 200 | 6 | 0.8 | 50 |
| | 60 | 5e-3 | 4 | 0.8 | 200 | 4 | 0.8 | 50 | 5e-3 | 2 | 0.8 | 200 | 4 | 0.8 | 50 |
| Citeseer | 20 | 5e-3 | 5 | 0.9 | 150 | 5 | 0.8 | 50 | 5e-3 | 3 | 0.85 | 150 | 5 | 0.8 | 50 |
| | 40 | 5e-3 | 2 | 0.85 | 150 | 3 | 0.8 | 50 | 5e-3 | 2 | 0.8 | 150 | 3 | 0.8 | 50 |
| | 60 | 5e-3 | 2 | 0.8 | 150 | 2 | 0.8 | 50 | 5e-3 | 6 | 0.8 | 150 | 2 | 0.8 | 50 |
| Pubmed | 20 | 5e-3 | 6 | 0.8 | 100 | 2 | 0.85 | 50 | 5e-3 | 2 | 0.8 | 100 | 2 | 0.85 | 50 |
| | 40 | 5e-3 | 4 | 0.8 | 100 | 2 | 0.85 | 50 | 5e-3 | 2 | 0.8 | 100 | 2 | 0.85 | 50 |
| | 60 | 5e-3 | 3 | 0.8 | 100 | 3 | 0.85 | 50 | 5e-3 | 2 | 0.85 | 100 | 2 | 0.85 | 50 |
| CoraFull | 20 | 0.03 | 4 | 0.85 | 500 | 3 | 0.95 | 50 | 0.03 | 5 | 0.95 | 500 | 3 | 0.95 | 50 |
| | 40 | 0.03 | 4 | 0.99 | 500 | 4 | 0.99 | 25 | 0.03 | 2 | 0.95 | 500 | 4 | 0.99 | 25 |
| | 60 | 0.03 | 5 | 0.9 | 500 | 2 | 0.95 | 25 | 0.03 | 2 | 0.95 | 500 | 2 | 0.95 | 25 |

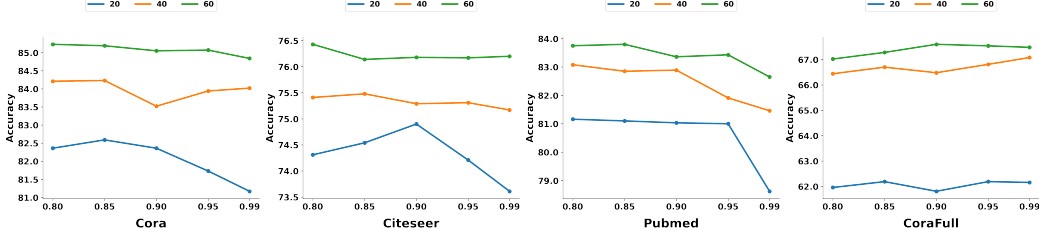

Figure 5: The accuracy changing trends on GCN w.r.t the threshold $\tau$

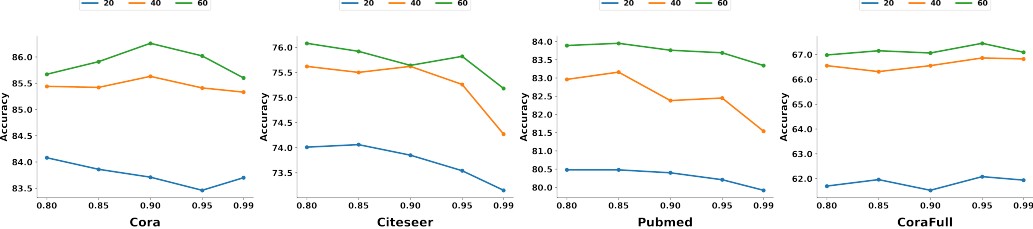

Figure 6: The accuracy changing trends on GAT w.r.t the threshold $\tau$

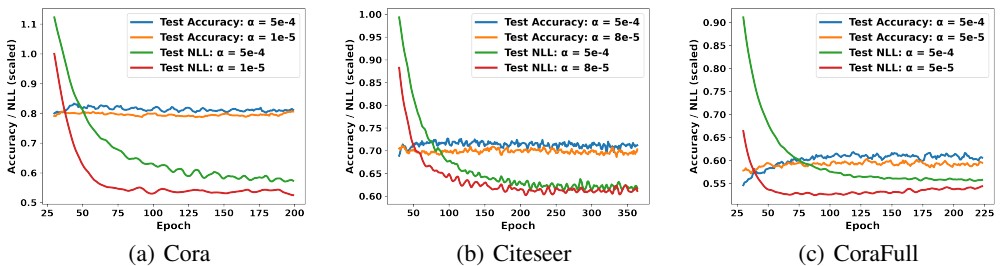

|     | (a) Cora | (b) Citeseer | (c) CoraFull |
|-----|----------|--------------|--------------|

Figure 7: The changing trends of test accuracy and NLL of GCN w.r.t epoch on three datasets.

## D  Why GCNs are poorly calibrated

In this section, we focus on the reason why GNNs are poorly calibrated. It is inspired by the observation in [12] that modern neural networks can overfit to NLL without overfitting to the accuracy. Since NLL can be used to measure model calibration as mentioned in Section 3.3, [12] gives an explanation of miscalibration: modern neural networks achieve better classification accuracy at the expense of well-calibrated probabilities. Similarly, we conduct an experiment to explore the relationship between NLL and the accuracy of GNNs.

We take the representative GCN as the example and apply it to Cora, Citeseer, CoraFull with label rate $L/C = 20$. We employ the validation set for early stopping in training with a window size of 100 and carefully tune the parameter $\alpha$ for weight decay to obtain the best accuracy and NLL on the test set respectively. Other parameters follow [16]. The changing trends of accuracy and NLL with respect to epoch are shown in Fig. 7, where NLL is scaled by a constant to fit the figure. Intuitively, GCN should achieve the best accuracy when NLL is the lowest. However, we find that GCN does not achieve the best performance when NLL is the lowest. Taking the Cora dataset in Fig. 7(a) as an example, we find that GCN achieves the best accuracy when $\alpha = 5e - 4$ but NLL still under-fits at this time, i.e., NLL has not achieved the lowest value. If we tune $\alpha$ to be $1e - 5$, NLL generally achieves the best result, i.e., GCN is better calibrated at this time, while accuracy drops from 81.5% to 79.5%. This gives an explanation of miscalibration for GNNs: GNNs learn better classification accuracy at the expense of well-modeled probabilities, i.e., GNNs *under-fit to NLL without under-fitting to accuracy*.

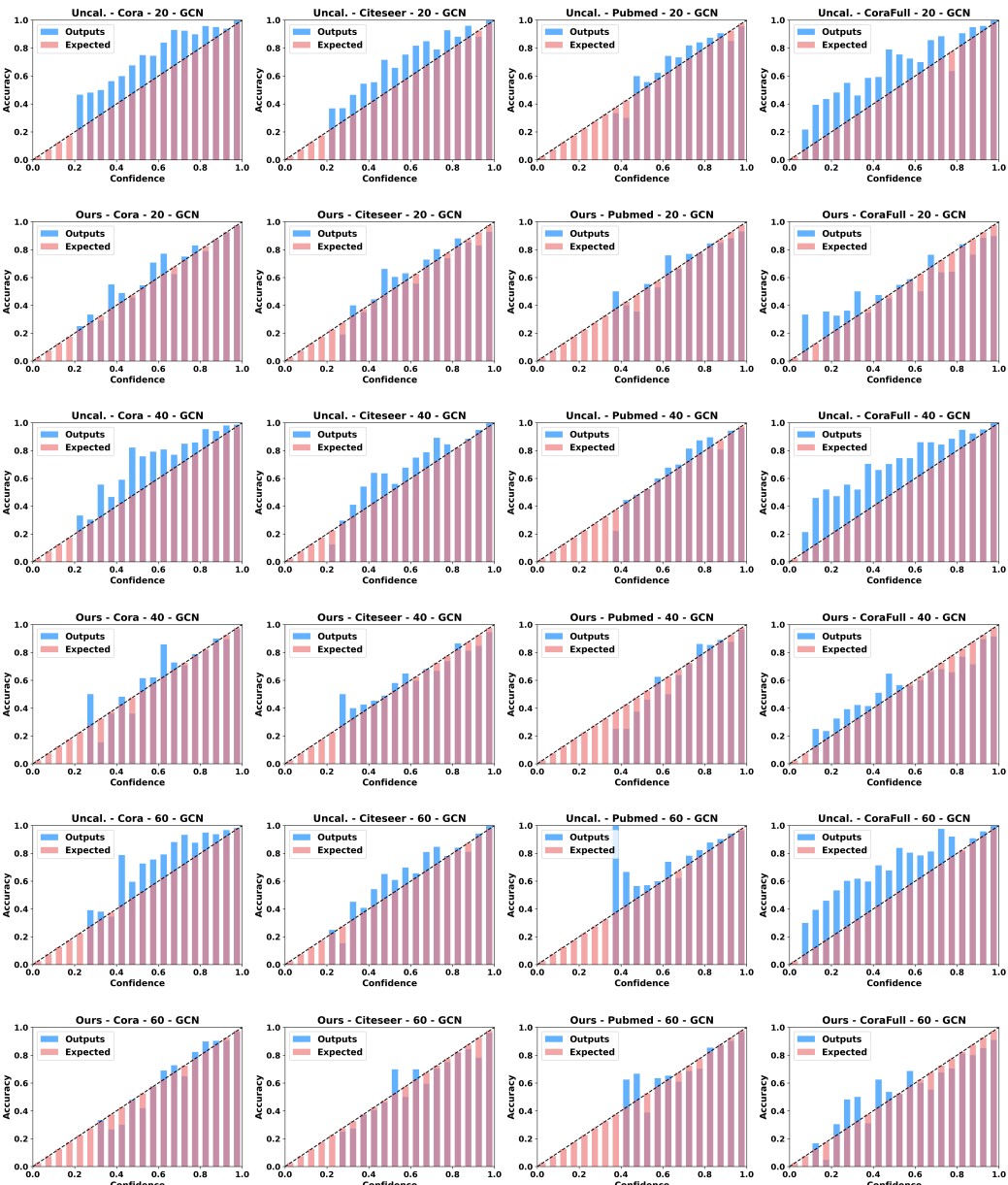

Figure 8: Reliability diagrams for GCN before (odd rows) and after (even rows) calibration.

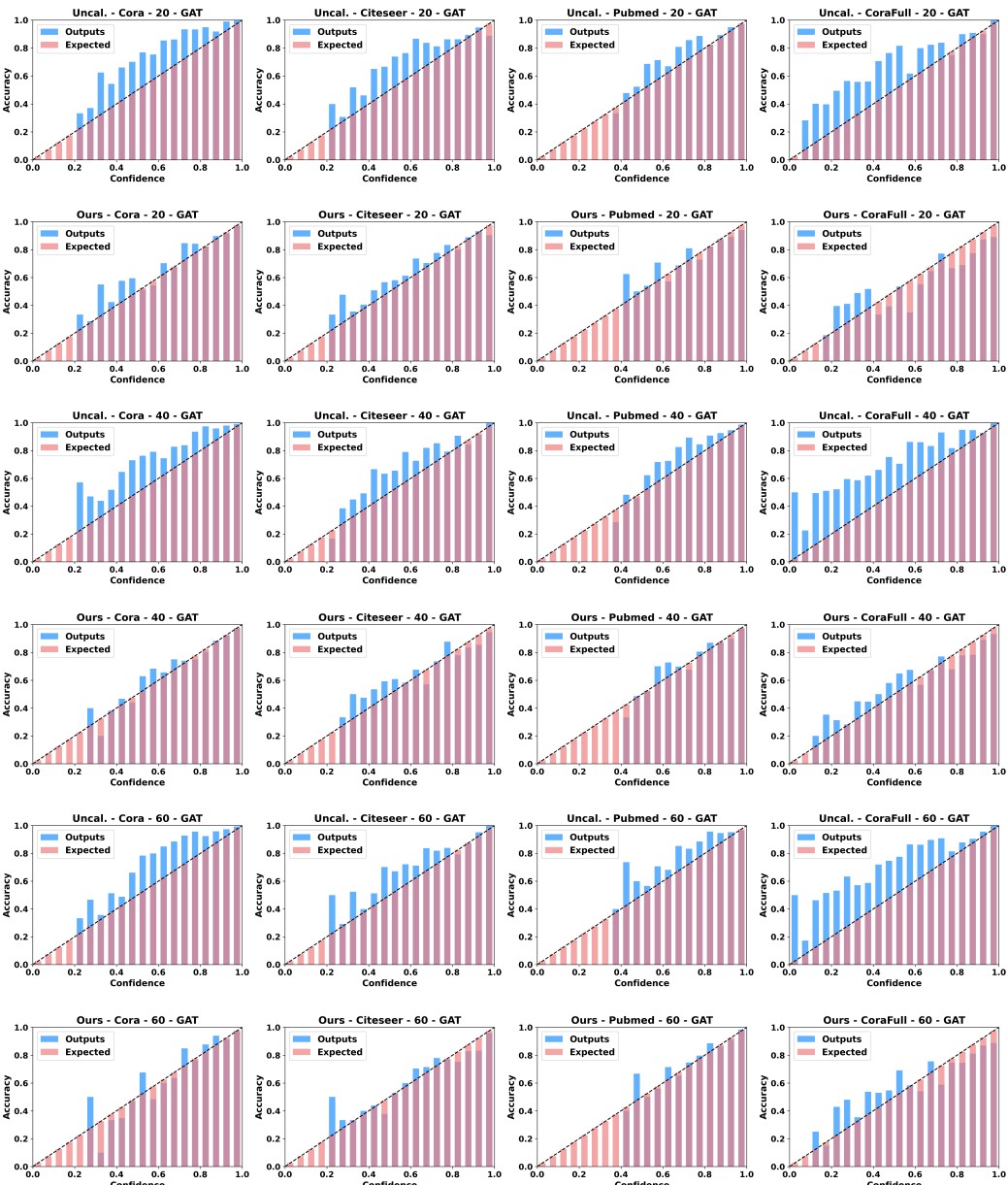

Figure 9: Reliability diagrams for GAT before (odd rows) and after (even rows) calibration.

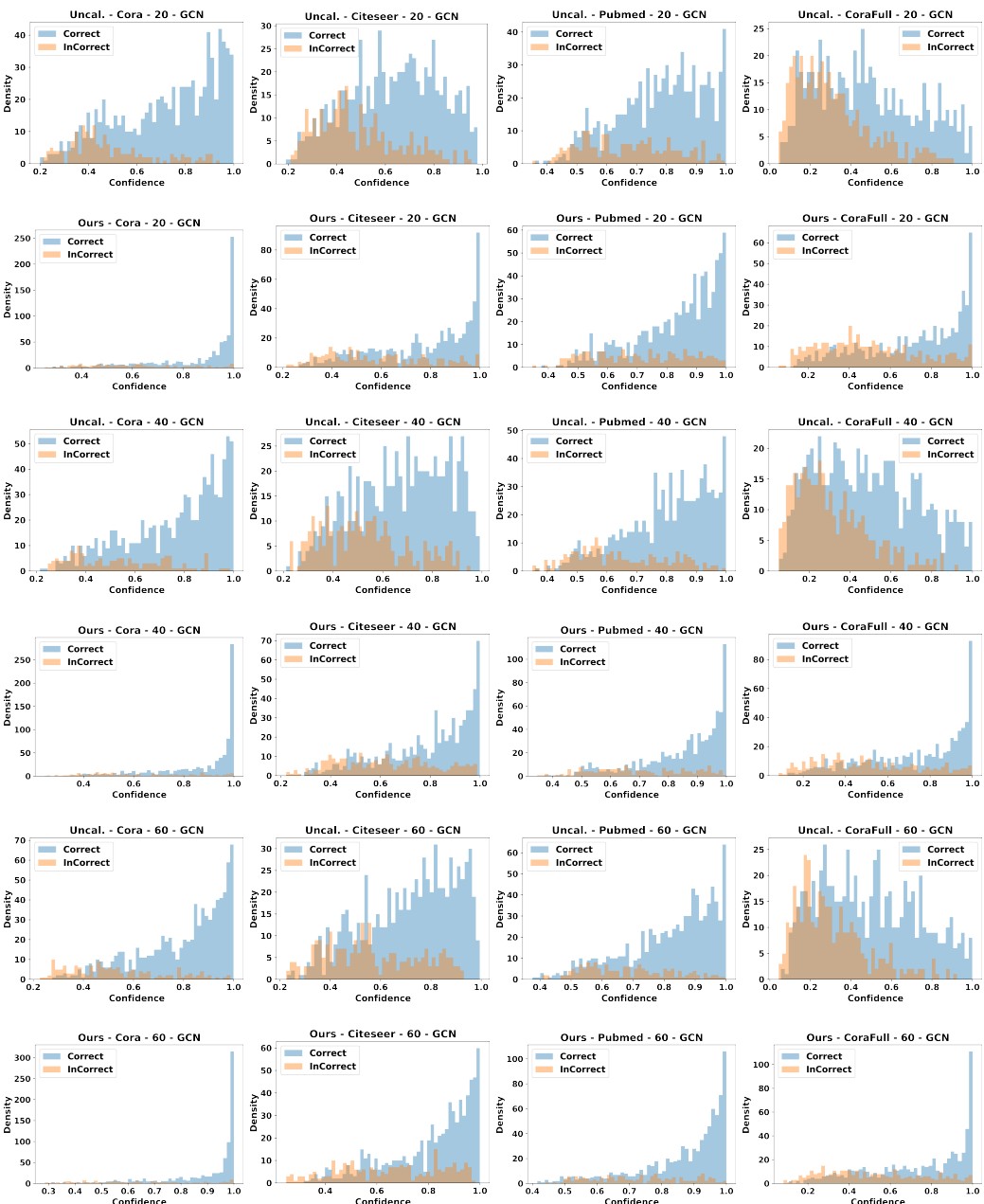

Figure 10: Confidence distribution before (odd rows) and after (even rows) calibration on GCN.

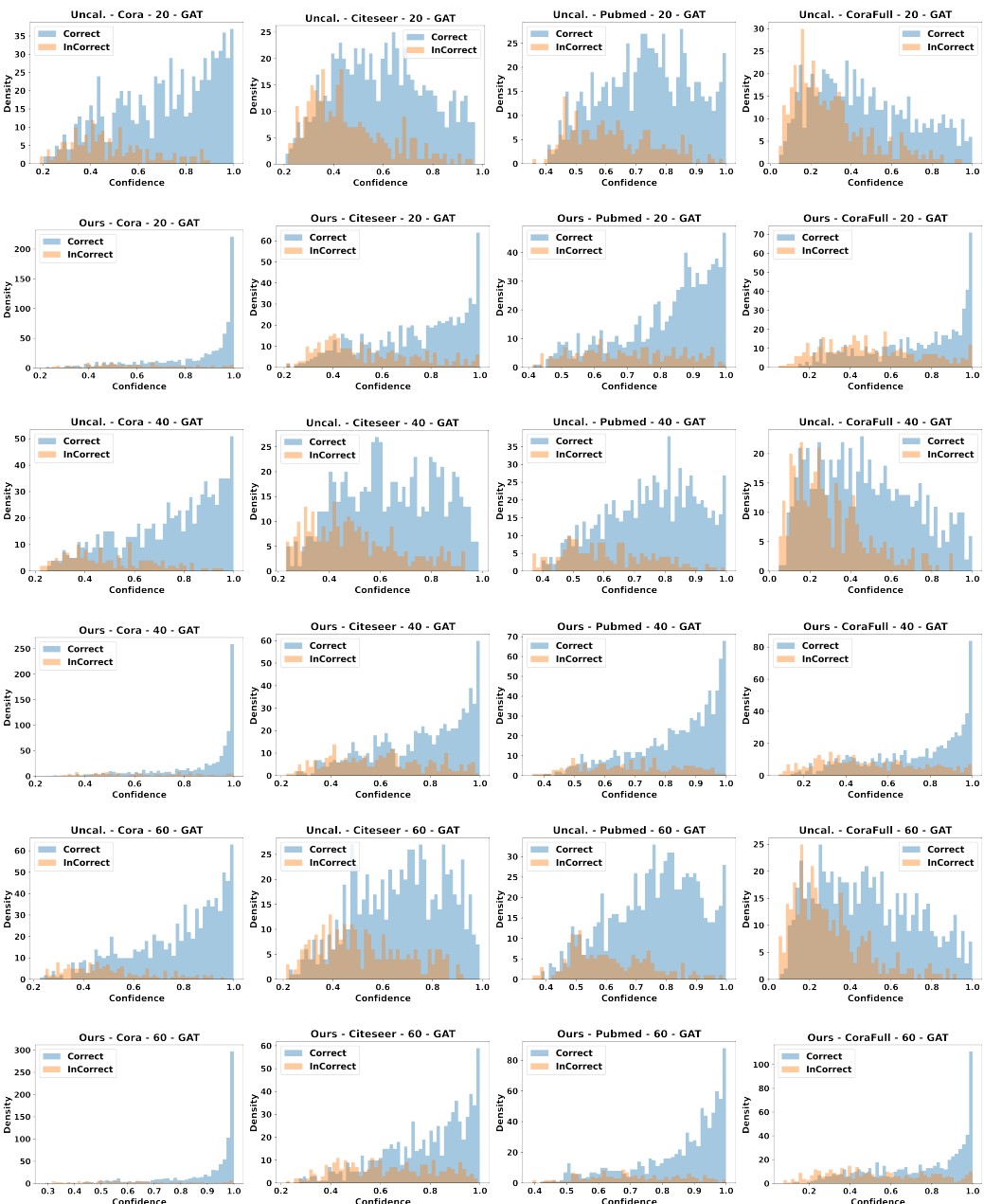

Figure 11: Confidence distribution before (odd rows) and after (even rows) calibration on GAT.

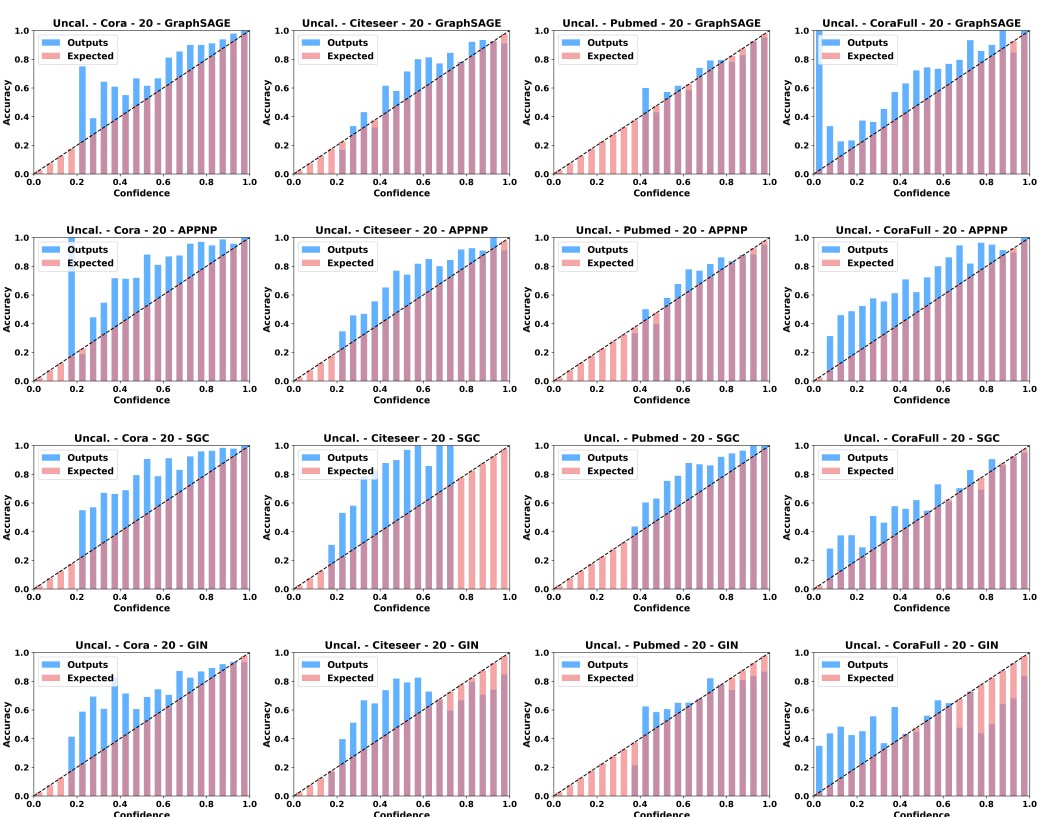

Figure 12: Reliability diagrams for GraphSAGE, APPNP, SGC, GIN with label rate to be 20.

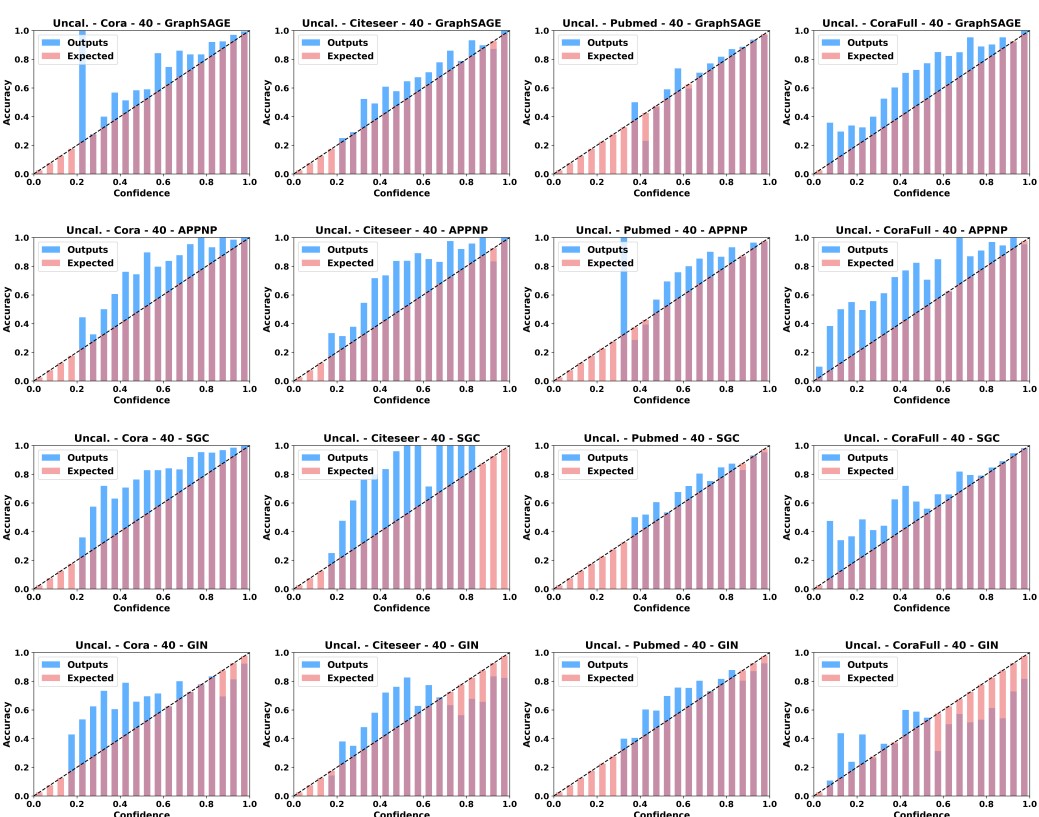

Figure 13: Reliability diagrams for GraphSAGE, APPNP, SGC, GIN with label rate to be 40.

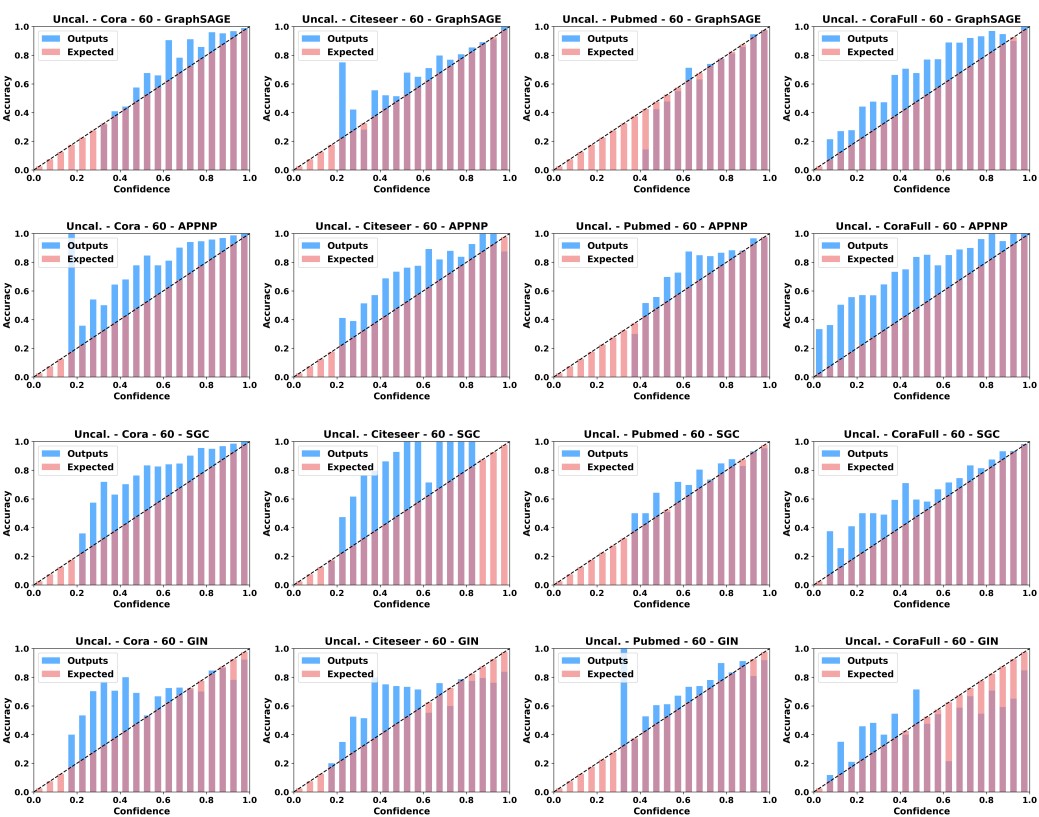

Figure 14: Reliability diagrams for GraphSAGE, APPNP, SGC, GIN with label rate to be 60.