# OpenReview forum: "Be Confident! Towards Trustworthy Graph Neural Networks via Confidence Calibration"
_NeurIPS.cc/2021/Conference — NeurIPS 2021 Poster_

### Official Review · Reviewer_4YpW · 2021-07-14

**Rating:** 6
**Confidence:** 4

**Summary:**

This paper aims to propose a novel graph neural network model that is trustworthy via confidence calibration. Graph neural networks (GNNs) attract many researchers’ attention, which has achieved excellent achievements. However, can we trust the predicted results of GNNs? Inspired by previous research works showing that model deep neural networks are over-confident with predictions, the authors study the confidence calibration of GNNs. Through many experiments and analyses, they find that the traditional GNNs are under-confident. This observation is different from previous research works related to the confidence calibration of deep neural networks. Thus, how to do the confidence calibration for GNNs is important and highly desired. The authors propose a new calibration GNN model (CaGCN) to learn the calibration function for improving the confidence of GNNs. Specifically, they first validate the confidence distribution in a graph owns homophily characteristics. Then, they transform logits of GNNs to the calibrated confidence of each node. Owing to GNNs is used to deliver confidential information among different nodes during the transformation process, the process preserves the order between classes. Moreover, they apply the CaGCN to a self-training framework, which generates trustworthy pseudo labels for unlabeled data to improve model performance. Finally, the authors conduct extensive experiments to validate the effectiveness from two perspectives: calibration and accuracy.

**Limitations And Societal Impact:**

There are some limitations in this paper: (1). The authors think that the predictive performance of GNNs is under-confident via analyzing the predictive performance of GNN and GAT. But I think the authors should test more graph neural networks to validate the conclusion. (2). The authors utilize GCNs to implement the confidence propagation part. They think the GCNs can preserve topology information and non-linear relations in graphs. But I think the motivation is weak. The authors should provide stronger reasons to support their model choice. For instance, can we utilize other non-linear graph models to capture these characteristics? (3). Figure 3 cannot reflect the confidence propagation, I suggest the authors replot this figure. (4). The authors can provide more application scenarios to highlight the social impact of the research.

**Main Review:**

Graph neural networks (GNNs) are popular in many domains. But can we trust the predicted result of GNNs? This research problem is interesting and useful. If we have strong and trustworthy GNNs, we can apply these GNNs to many real-world applications, especially safety-critical fields. Thus, from this perspective, we can find that the research topic of this paper is novel and important.

In addition, from the presentation perspective, this paper is well-organized and readers can follow the logic of the paper easily. Specifically, this paper has two steps: (1) finding the under-confident of GNNs, (2) proposing a calibrated GNNs (CaGNN) to make the predicted results more trustworthy. For the first step, the authors find the under-confident of GNNs via experiments, visualizations, and analyses. For the second step, the authors provide detailed explanations.

Moreover, from the experimental perspective, there are two advantages in this paper: (1) the authors conduct extensive experimental results to validate the effectiveness of their framework. Through these experiments, we find that the authors’ framework is not only more stable but also more accurate compared with other baseline models; (2) for each experiment, the authors provide detailed experimental settings including parameter setting, dataset, and so on, which is beneficial for other readers to reproduce experimental results.

**Time Spent Reviewing:**

8

---

> ### Author Response · Authors · 2021-08-10
> **Response**
>
> We sincerely thank the Reviewer for spending time and providing valuable feedback. We appreciate all of your suggestions and we have addressed all your questions below by providing our responses as well as our additional experimental results.
>
> 1. > The authors think that the predictive performance of GNNs is under-confident via analyzing the predictive performance of GNN and GAT. But I think the authors should test more graph neural networks to validate the conclusion.
>
> **Response:** Thank you for the suggestion. We employ another four representative GNNs (GraphSAGE [1], APPNP [2], SGC [3], GIN [4]) to conduct the experiments on all the datasets with 20, 40, 60 label rates, so as to more comprehensively analyze the confidence distribution of GNNs. We still illustrate the results using Reliability Diagrams as in Section 2 in our paper. Because we cannot insert figures into our response here, for better visual presentation, we report these figures at an anonymous URL https://imgur.com/a/fphHBul.
>
> As shown in these figures, the x-axis of digrams is the confidence in 20 bins of equal size while the y-axis is the average accuracy in each bin. The blue represents the classification accuracy of different models (GraphSAGE, APPNP, SGC, GIN) while the red is our expectation. From the reliability diagrams we can observe that in almost all the datasets, the average accuracy of most bins is higher than the average confidence, which means these models are under-confident, verifying the conclusion in the paper again.
>
>    We will add these figures and the analysis to our appendix in the revision.
>
> 2. > The authors utilize GCNs to implement the confidence propagation part. They think the GCNs can preserve topology information and non-linear relations in graphs. But I think the motivation is weak. The authors should provide stronger reasons to support their model choice. For instance, can we utilize other non-linear graph models to capture these characteristics?
>
> **Response:** We choose GCN as our confidence calibration model based on two facts: one is that, as shown in Section 3.1, we first verify that if a classification model is well-calibrated, the total variation (TV) of confidence will be smaller and the confidence distribution has homophily property. This motivates us that we should find a function which is able to smooth the confidence along the network topology. Then it is well known that GCN is able to smooth the node signals along the network topology to decrease the TV of signals. The other is that GCN can capture the homophily property, which has been proved in [5]. Both the empirical study in Section 3.1 and the theoretical analysis in [5] implies that GCN can be considered as the basic calibration model.
>
> Moreover, [5] has also theoretically proved that the propagation mechanism of most existing GNNs (GCN, SGC, APPNP, JKNet, DAGNN, et al.) can be reformulated as one unified optimization framework, including one term implicitly smoothing the features along the network topology and capturing the homophily property. Based on these analyses, it makes sense that these GNNs can be probably considered as the calibration model for CaGCN here.  Thank you again for your comment and we think it is worth making further study in this direction.
>
> 3. > Figure 3 cannot reflect the confidence propagation, I suggest the authors replot this figure.
>
> **Response:** Thanks for your suggestion. We replot this figure about confidence propagation for a clearer explanation. Because we cannot insert figures into our response here, we report it at an anonymous URL https://imgur.com/BJYm1QT and we will replace Fig. 3 with this figure in the new version. The explanation for this figure is still the same as stated in Section 3.1.
>
> 4. > The authors can provide more application scenarios to highlight the social impact of the research.
>
> **Response:** Thank you for your suggestion. Current efforts on advancing GNNs mostly focus on classification accuracy. However, when deploying GNNs to real-world applications, especially safety-critical fields, whether the results of GNNs are trustworthy is another important factor. The demands for a trustworthy model are universal and extensive such as in the field of disease prediction [6], financial fraud detection [7], traffic states prediction [8] and object detection [9] for autonomous driving, where estimating the true probability of getting a correct prediction is necessary.
>
> Firstly, we take the disease prediction [6] as an example, where GNNs are utilized to encode the information of different symptoms, users and diseases. In this scenario, accurately and comprehensively predicting diseases at an early stage will help patients receive prevention treatments in a timely manner. Otherwise, the misdiagnosis and missed diagnosis will endanger the health of patients. Therefore, a trustworthy model is urgently needed. Our CaGCN can make a trustworthy prediction based on its confidence, and as a result, decrease the risk of misdiagnosis and missed diagnosis.
>
> Secondly, we take financial fraud detection [7] as another example, where GNNs are utilized to capture rich interactions in the financial scenarios to predict whether an entity will be involved in fraud or not in the future. Accurately detecting financial fraud will enhance the security of both the users and the service providers. However, existing GNNs are under-confident, leading that many fraud cases are neglected because of the low confidence. Our trustworthy CaGCN can also decrease the risk in this scenario.
>
> We will add these scenarios to our paper in the revision and we hope the above analysis can clarify the social impact of our work. Thanks.
>
>    [1] Hamilton W L, Ying R, Leskovec J. Inductive representation learning on large graphs[C]//Proceedings of the 31st International Conference on Neural Information Processing Systems. 2017: 1025-1035.
>
>    [2] Klicpera J, Bojchevski A, Günnemann S. Predict then propagate: Graph neural networks meet personalized pagerank[J]. arXiv preprint arXiv:1810.05997, 2018.
>
>    [3] Wu F, Souza A, Zhang T, et al. Simplifying graph convolutional networks[C]//International conference on machine learning. PMLR, 2019: 6861-6871.
>
>    [4] Xu K, Hu W, Leskovec J, et al. How powerful are graph neural networks?[J]. arXiv preprint arXiv:1810.00826, 2018.
>
>    [5] Zhu M, Wang X, Shi C, et al. Interpreting and unifying graph neural networks with an optimization framework[C]//Proceedings of the Web Conference 2021. 2021: 1215-1226.
>
>    [6] Sun Z, Yin H, Chen H, et al. Disease Prediction via Graph Neural Networks[J]. IEEE Journal of Biomedical and Health Informatics, 2020, 25(3): 818-826.
>
>    [7] Wang D, Lin J, Cui P, et al. A semi-supervised graph attentive network for financial fraud detection[C]//2019 IEEE International Conference on Data Mining (ICDM). IEEE, 2019: 598-607.
>
>    [8] Cui Z, Henrickson K, Ke R, et al. Traffic graph convolutional recurrent neural network: A deep learning framework for network-scale traffic learning and forecasting[J]. IEEE Transactions on Intelligent Transportation Systems, 2019, 21(11): 4883-4894.
>
>    [9] Gu J, Hu H, Wang L, et al. Learning region features for object detection[C]//Proceedings of the european conference on computer vision (ECCV). 2018: 381-395.

---

### Official Review · Reviewer_TDBE · 2021-07-15

**Rating:** 5
**Confidence:** 5

**Summary:**

This paper analyzes the GNN prediction confidence, and reveals that the prediction confidence made by GNN is lower than the accuracy derived from the prediction confidence values, i.e., GNNs are under-confident.  This is different from the previously analyzed modern neural network models, which are often over-confident on the prediction (the prediction accuracy is lower than its confidence). To make the confidence values usable to reflect the true prediction confidence in safety-critical applications, a confidence calibration mechanism is proposed to make the confidence value be exactly equal to the true probability of getting a correct prediction for every node. The calibration follows the classic temperature scaling method, which has been used in the calibration of modern neural networks (ref [10]).

**Main Review:**

It is a novel study on the confidence calibration of GNN models, given that the previous work focused on the calibration of modern neural networks (ref [10]).  However, the paper has to improve on several aspects. First, the description of the model has unclarified facts. Second, the evaluation should be conducted with statistical test to show the effectiveness of the proposed solution. Last, some writing inconsistency should be addressed.  More details can be found below.

In line 175, “, the order between classes of v_i and v’_i is not the same, implying that after calibration by CaGCN, the accuracy of original GCN cannot be preserved. ” But softmax is a monotonic function. The order between classes of v_i and v’_i should be the same.

Comparing to the classic temperature scaling method in the calibration of modern neural networks (ref [10]), the calibration in Eq(5) introduce different temperature t_i for different nodes. This is an interesting idea. As shown in figure 4, the weights W in Eq(5) are shared with the weights in Eq(3) in GCN for getting output V. In this way, t_i is in fact from t_i = \sigma+(v_i).  However, v_i is a vector. A softplus activation function applied on a vector v_i cannot result in a scalar t_i. Clarification is needed.

Regarding the evaluation results, Table 3 reports the node classification accuracy and the variance (over 10 independent runs) of different models. The variance is quite large and the accuracy difference is small. Statistical test should be done to show if there is a significant improvement between CaGCN-st and baselines.  Table 2 doesn’t show the variance of ECE of different models. Given the small difference between CaGCN and other baselines, it would be necessary to present the variance and run statistical test as well.

Writing to correct:
1.	In Definition 1,  label index in Y is from 0 to K.  But z_i is a vector with elements from 1 to K (should be from 0 to K). Please check also in many other places, the index is sometimes from 1 to K, and sometimes from 0 to K.
2.	Same for the index of node i. The index below Eq (2) is from 0 to N, for the total N nodes above Definition 1.  In Eq (2), i is indexed rom 1 to N. But in some other places, i is indexed from 0 to N.
3.  a l-layer GCN    etc


**Time Spent Reviewing:**

6 hours

---

> ### Author Response · Authors · 2021-08-10
> **Response (1/2)**
>
> We sincerely thank the Reviewer for your careful reading and your correction of our mistakes. We feel very sorry for the negligence in the description of the model. We will correct them in our paper. We would like to address the concerns of the Reviewer by providing our responses as well as our additional experimental results.
>
> 1. > In line 175, “ the order between classes of $v_i$ and $v'_i$​​ is not the same, implying that after calibration by CaGCN, the accuracy of original GCN cannot be preserved. ” But softmax is a monotonic function. The order between classes of v_i and v’_i should be the same.
> **Response:** Actually, as shown in Line 136 and Eq.3, CaGCN is another $l$​​​​-layer GCN, implying that softmax operation is not the only step in CaGCN. Instead, CaGCN contains two main steps: one is the confidence propagation to get the calibrated logits $\mathbf{V'}$​​​​​ given the output $\mathbf{V}$​​​​​ of the original GCN model, and the second step is normalization utilizing softmax operation. We agree that the second step, i.e., the softmax, will not change the order, but the first step will. More specifically, as can be seen in Eq.3, given the logits $\mathbf{V}$​​​​​ of original GCN, CaGCN aggregates and transforms $\mathbf{V}$​​​​​ to obtain the logits $\mathbf{V'}$​​​ after calibration via $\mathbf{V'}=\mathbf{A}\sigma(\cdots\mathbf{A}\sigma(\mathbf{A}\mathbf{V}\mathbf{W}^{(1)})\mathbf{W}^{(2)\cdots})\mathbf{W}^{(l)}$​​​. We denote the prediction by $\hat{y}\_i = \arg\max_k\sigma_{SM}(\mathbf{v'}\_{i,k}) $​ for each node $i$​​​, where $\sigma_{SM}(\cdot)$​​​ refers to the softmax operation. It should be noted that the operation $\mathbf{A}\mathbf{V}\mathbf{W}^{(1)}$​​​ in Eq.3 is not monotonic, that is to say, the transformation from $\mathbf{V}$​​​ to $\mathbf{V'}$​​​ is not monotonic, leading that the prediction $\hat{y}_i$​​​​​​ for original GCN and CaGCN is not the same.
>
> 2. > Comparing to the classic temperature scaling method in the calibration of modern neural networks (ref [10]), the calibration in Eq(5) introduces different temperature t_i for different nodes. This is an interesting idea. As shown in figure 4, the weights W in Eq(5) are shared with the weights in Eq(3) in GCN for getting output V. In this way, t_i is in fact from t_i = \sigma+(v_i). However, v_i is a vector. A softplus activation function applied on a vector v_i cannot result in a scalar t_i. Clarification is needed.
>
> **Response:** We are sorry for this unclear statement, which causes your confusion. Actually, the weights W in Eq.5 are not the weights in Eq.3. CaGCN in Eq.3 is our basic confidence calibration model, which is used to verify that GCN can be applied to calibrate the confidence. However, as demonstrated in Line 174-176, despite the CaGCN in Eq.3 can calibrate the confidence, it cannot preserve the accuracy. As a result, we further propose an improved CaGCN in Eq.5 which can preserve the accuracy as well. This implies that CaGCN in Eq.5 is not the one in Eq.3. More specifically,  the dimension of the weights $\mathbf{W}^{(l)}$​​​ in Eq.3 is $\cdot\times K$​​​ as the output of CaGCN in Eq.3 is $\mathbf{Z}\in\mathbb{R}^{N\times K}=\sigma_{SM}(\mathbf{A}\sigma(\cdots\mathbf{A}\sigma(\mathbf{A}\mathbf{V}\mathbf{W}^{(1)})\mathbf{W}^{(2)})\mathbf{W}^{(l)})$​​​, where $N$​​​ and $K$​​​ are the number of nodes and classes respectively, and we denote the softmax operation by $\sigma_{SM}(\cdot)$​​​ for simplicity. On the other hand, the dimension of the weights $\mathbf{W}^{(l)}$​​​ in Eq.5 is $\cdot\times1$​​​ as the output of CaGCN in Eq.5 is $\mathbf{t}\in \mathbb{R}^N=\sigma^+(\mathbf{A}\sigma(\cdots\mathbf{A}\sigma(\mathbf{A\mathbf{V}\mathbf{W}^{(1)}})\mathbf{W}^{(2)})\mathbf{W}^{(l)})$​​​, where $\sigma^+(\mathbf{x})=\log (1+ \exp (\mathbf{x}))$​​​ is a softplus operation for each dimension of the vector $\mathbf{x}$​​​.
>
> Thank you for your careful reading and suggestions. We will revise the relevant parts carefully in the revision.

---

> ### Author Response · Authors · 2021-08-10
> **Response (2/2)**
>
> 3. > Regarding the evaluation results, Table 3 reports the node classification accuracy and the variance (over 10 independent runs) of different models. The variance is quite large and the accuracy difference is small. Statistical test should be done to show if there is a significant improvement between CaGCN-st and baselines. Table 2 doesn’t show the variance of ECE of different models. Given the small difference between CaGCN and other baselines, it would be necessary to present the variance and run statistical test as well.
>
> **Response:** Thank you. Based on your suggestions, we provide the standard deviation of ECE and the results of statistical test for confidence calibration and self-training. The statistical test is done between our method and the second-best method (MS on Cora, Citeseer, Pubmed and TS on CoraFull for confidence calibration respectively, and TS-st for self-training). We report these results in Table 1 and Table 2 below. For each result, the subscript refers to the standard deviation ($\times 10^{-3}$ for confidence calibration) while the superscript refers to the results of paired t-test ( * for 0.05 level and ** for  0.01 level). (-) denotes this method cannot converge to a meaningful result and the bold denotes the best result.
>
> Table 1 reports the calibration results evaluated by ECE on different models and citation networks of various label rate (L/C) before and after calibration. From Table 1 we can observe the results are statistically significantly better at the * 0.05 level and ** 0.01 level. Table 2 reports the node classification accuracy and its standard deviation (%) on CaGCN-st and TS-st. From Table 2, we can also observe that the results are statistically significantly better at the * 0.05 level. We will add these results to our paper in the revision.
>
> Table 1: Results of confidence calibration. Uncal., TS, MS represent the uncalibrated model, temperature scaling, matrix scaling respectively.
>
> | Dataset  | L/C  |      GCN       |                         |                         |                              |      |      GAT       |                |                |                              |
> | :------: | :--: | :------------: | :---------------------: | :---------------------: | :--------------------------: | ---- | :------------: | :------------: | :------------: | :--------------------------: |
> |          |      |     Uncal.     |           TS            |           MS            |            CaGCN             |      |     Uncal.     |       TS       |       MS       |            CaGCN             |
> |   Cora   |  20  | $0.1347_{6.3}$​ |     $0.0488_{5.5}$​​      |     $0.0414_{5.7}$​​      |   $\mathbf{0.0401}_{6.7}$​​​    |      | $0.1558_{8.9}$​​ | $0.0717_{9.8}$​​ | $0.0544_{9.4}$​​ | $\mathbf{0.0450}_{5.6}^{**}$​​​​ |
> |          |  40  | $0.1134_{4.7}$​ |     $0.0417_{7.2}$​      | $\mathbf{0.0372}_{4.6}$​​ |        $0.0407_{5.4}$​        |      | $0.1340_{5.4}$​ | $0.0485_{7.7}$​ | $0.0491_{6.0}$​ | $\mathbf{0.0365}_{5.6}^{**}$​​​ |
> |          |  60  | $0.0937_{4.9}$ | $\mathbf{0.0355}_{5.4}$​​ |     $0.0364_{6.1}$​      |        $0.0376_{4.4}$​        |      | $0.1201_{3.3}$​ | $0.0393_{6.1}$​ | $0.0411_{5.3}$​ | $\mathbf{0.0313}_{3.2}^{**}$​​ |
> | Citeseer |  20  | $0.1248_{7.1}$​ |     $0.0641_{8.7}$​      |     $0.0644_{3.7}$​      | $\mathbf{0.0595}_{7.2}^{*}$​​​  |      | $0.1534_{5.0}$​ | $0.0916_{8.7}$​ | $0.0633_{9.8}$​ |   $\mathbf{0.0572}_{6.8}$​​    |
> |          |  40  | $0.0957_{7.7}$​ |     $0.0601_{4.2}$​      | $\mathbf{0.0538}_{5.7}$​​ |        $0.0545_{5.5}$​        |      | $0.1252_{8.7}$​ | $0.0797_{3.1}$​ | $0.0590_{5.4}$​ | $\mathbf{0.0532}_{5.4}^{*}$​​​  |
> |          |  60  | $0.0806_{6.4}$​ |     $0.0559_{5.0}$​      | $\mathbf{0.0521}_{6.4}$​​ |        $0.0546_{3.4}$​        |      | $0.1090_{5.9}$​ | $0.0648_{7.1}$​ | $0.0519_{9.1}$​ |   $\mathbf{0.0525}_{7.6}$​​    |
> |  Pubmed  |  20  | $0.0586_{7.7}$​ |     $0.0541_{3.8}$​      |     $0.0476_{4.2}$​      |  $\mathbf{0.0405}_{6.0}^*$​​   |      | $0.0835_{3.1}$​ | $0.0656_{4.6}$​ | $0.0501_{3.7}$​ | $\mathbf{0.0356}_{6.3}^{**}$​​​ |
> |          |  40  | $0.0444_{5.5}$​ |     $0.0446_{6.3}$​      |     $0.0436_{6.3}$​​      | $\mathbf{0.0402}_{4.0}^{*}$​​​  |      | $0.0869_{4.6}$​ | $0.0658_{6.5}$​ | $0.0539_{6.0}$​ | $\mathbf{0.0308}_{5.4}^{**}$​​​ |
> |          |  60  | $0.0445_{9.7}$​ |     $0.0367_{6.0}$​      |     $0.0318_{6.4}$​      |   $\mathbf{0.0311}_{4.8}$​​    |      | $0.0993_{4.1}$​ | $0.0669_{6.3}$​ | $0.0483_{5.7}$​ | $\mathbf{0.0308}_{5.2}^{**}$​​​ |
> | CoraFull |  20  | $0.1986_{6.1}$​ |     $0.1013_{6.1}$​      |            -            | $\mathbf{0.0776}_{6.4}^{**}$​​​ |      | $0.2119_{3.6}$​ | $0.1101_{5.1}$​ |       -        | $\mathbf{0.0788}_{6.0}^{**}$​​ |
> |          |  40  | $0.2321_{5.4}$​ |     $0.1117_{6.5}$​      |            -            | $\mathbf{0.0701}_{3.9}^{**}$​​​ |      | $0.2438_{4.2}$​ | $0.1133_{8.3}$​ |       -        | $\mathbf{0.0738}_{4.8}^{**}$​​​ |
> |          |  60  | $0.2337_{4.0}$​ |     $0.0981_{3.8}$​      |            -            | $\mathbf{0.0768}_{3.4}^{**}$​​​ |      | $0.2497_{1.8}$​ | $0.1133_{5.2}$​ |       -        | $\mathbf{0.0849}_{6.9}^{**}$​​​ |
>
> Table 2: Node classification accuracy and its standard deviation (%) on CaGCN-st and TS-st.
>
> | Dataset  | L/C  |          TS-st          |           CaGCN-st           |
> | :------: | :--: | :---------------------: | :--------------------------: |
> |   Cora   |  20  |     $82.68_{0.20}$      |  $\mathbf{83.11}_{0.52}^*$​​   |
> |          |  40  | $\mathbf{84.44}_{0.35}$​ |        $84.37_{0.38}$        |
> |          |  60  |     $85.60_{0.24}$      |   $\mathbf{85.79}_{0.27}$​    |
> | Citeseer |  20  |     $74.20_{0.24}$      | $\mathbf{74.90}_{0.40}^{**}$​​ |
> |          |  40  | $\mathbf{75.62}_{0.19}$​ |        $75.48_{0.50}$        |
> |          |  60  |     $75.87_{0.24}$      | $\mathbf{76.43}_{0.20}^{**}$​​ |
> |  Pubmed  |  20  |     $80.95_{0.18}$      | $\mathbf{81.16}_{0.36}^{*}$​​  |
> |          |  40  |     $82.28_{0.39}$      |  $\mathbf{83.08}_{0.21}^*$​​   |
> |          |  60  |     $83.26_{0.39}$      | $\mathbf{84.47}_{0.23}^{**}$​​ |
> | CoraFull |  20  |     $61.73_{0.41}$      |  $\mathbf{62.19}_{0.49}^*$​​   |
> |          |  40  |     $66.11_{0.60}$      |  $\mathbf{66.30}_{0.31}^*$​​   |
> |          |  60  |     $66.95_{0.45}$      |  $\mathbf{67.60}_{0.40}^*$   |

---

### Official Review · Reviewer_KZsj · 2021-07-24

**Rating:** 8
**Confidence:** 3

**Summary:**

This paper proposed a confidence calibration method for graph neural networks (GNNs) for semi-supervised node classification problems. Based on the existing researches such as temperature scaling, the authors attempt to evaluate the confidence problem of GNNs and find a unique phenomenon that GNNs including GCN and GAT are less confident in their predictions which seem to be constant to other neural networks. From this observation, they proposed a confidence calibration procedure being aware of the topology of a graph structure and prove the accuracy-preserving property of their method. They also provided an effective optimization procedure for training their objective. Intensive experiments demonstrated the improvements of their proposed method with regard ECE and classification accuracy.

**Limitations And Societal Impact:**

Yes

**Main Review:**

This is a well-written paper. The main question "will the current GNNs follow the same over-confident property as other neural networks?" of this paper is clear and interesting for not only me but other researchers. I enjoyed reading it.

Strong points:
1. The authors firstly tackled the confidence calibration problem of the GNNs and report under-confident phenomena of GNN that have not been reported in the context of the confidence calibration of other neural networks.
2. To fix the under-confidence problem of GNN, they proposed a new calibration GNN model using an idea based on the homophily property. Although its structure is a bit simple and having no significant difference from the existing GCN, they proved the accuracy-preserving property that assures that no decline is caused by their CaGCN.
3. Intensive experiments showed better performances of CaGCN from other confidence calibration methods such as TS and MS.

Weak points:
1. In l.131 of p. 4, they employed the total variation to assess the confidence distribution and argued that "We can find that the total
variation of confidence does decrease after temperature scaling, which verifies our assumption." It is bit confusing for me. I suppose that if a GCN is well-calibrated, its confidence distribution is also calibrated. And thus a TS operation may not able to decrease the TV. Could you clarify your arguments more?
2. In Table 3, the authors report the node classification accuracy for four datasets. Although the accuracy-preserving property is proved in the aforementioned section, I cannot understand why accuracy of the TS-st and CaGCN-st were improved from the baseline. Is it caused by the calibration procedure or just the increasing number of parameters and layers of baselines?
3. In the context of graph neural networks, node classification is one of the fashionable problems but is a sub-class of the GNN task. Is it possible to enlarge the coverage of their proposed method to the other GNN tasks?

**Time Spent Reviewing:**

5 hours

---

> ### Author Response · Authors · 2021-08-10
> **Response**
>
> We sincerely thank the Reviewer for all the comments and it is a great honor for us for your enjoying our paper. We have addressed all your questions below and hope they have clarified all confusion you had about our work.
>
> 1. > In l.131 of p. 4, they employed the total variation to assess the confidence distribution and argued that "We can find that the total variation of confidence does decrease after temperature scaling, which verifies our assumption." It is bit confusing for me. I suppose that if a GCN is well-calibrated, its confidence distribution is also calibrated. And thus a TS operation may not able to decrease the TV. Could you clarify your arguments more?
>
> **Response:** Sorry for our unclear statement, causing your confusion. We agree that if a GCN has been well-calibrated, temperature scaling (TS) may not be able to decrease the total variation (TV). Actually, here our experiment in Section 3.1 is not conducted on a well-calibrated GCN, but on an uncalibrated GCN, because our motivation is to examine how the total variation of confidence will change when a GCN is from uncalibrated to well-calibrated. We apply TS to an uncalibrated GCN to make it well-calibrated, then we can analyze the change of confidence distribution. We discover that compared with the total variation of confidence with uncalibrated GCN, the total variation of confidence with well-calibrated GCN (i.e., after TS) does decrease, which verifies our assumption that the ground-truth confidence distribution in a graph should have homophily property. We will revise it and make a clear explanation in our revision.
>
>
> 2. > In Table 3, the authors report the node classification accuracy for four datasets. Although the accuracy-preserving property is proved in the aforementioned section, I cannot understand why accuracy of the TS-st and CaGCN-st were improved from the baseline. Is it caused by the calibration procedure or just the increasing number of parameters and layers of baselines?
>
> **Response:** Actually, the accuracy-preserving property is proved for CaGCN model in Section 3.2, rather than CaGCN-st framework in Section 4. CaGCN is designed for confidence calibration, which can preserve the classification accuracy. However, CaGCN-st is a self-training framework with CaGCN. Specifically, we firstly employ CaGCN to calibrate the confidence given by the GCN model and then for those unlabeled nodes with high confidence, we can label them with pseudo labels and retrain the GCN model by adding the newly labeled nodes to the training set. After being calibrated, the confidence of most of correct predictions becomes higher while incorrect predictions are basically unchanged, as shown in Fig. 10 and  Fig. 11 in Appendix. As a result, CaGCN-st can adopt more correct predictions as pseudo labels compared with other self-training methods, achieving better performance. Therefore, the results of CaGCN-st, shown in Table 3, are improved. This experiment is used to demonstrate the benefit of confidence calibration for self-training.
>
> As for CaGCN, it is a confidence calibration model with the accuracy-preserving property, as a result, we conduct experiments on confidence calibration evaluated by ECE in Section 5.1 as well as NLL and Brier Score in Appendix C.1 to demonstrate the effectiveness.
>
> 3. > In the context of graph neural networks, node classification is one of the fashionable problems but is a sub-class of the GNN task. Is it possible to enlarge the coverage of their proposed method to the other GNN tasks?
>
> **Response:** The node classification is one of the most important tasks in many GNNs papers [1, 2]. Therefore, following their settings [1, 2], we use node classification as our main task. Here we would like to thank the reviewer very much for this valuable suggestion, which further inspire us. Generally, we think that our idea may still work on other graph tasks, but more studies need to be conducted.
>
> Firstly, for the graph classification task, we can still employ CaGCN to propagate the node confidence along the network topology, and get the final calibrated confidence for a graph by pooling the confidence of all the nodes. Secondly, for the link prediction task,  we can regard the link prediction as a binary classification problem and the output as the confidence. Considering that the ground-truth confidence distribution for nodes should have the homophily property as is shown in Section 3.1, edges are likely to have the same property as well. As a result, we can employ CaGCN to propagate the confidence between edges by regarding the edges as the nodes. However, more exploration still needs to be conducted for the homophily property of edges. Lastly, for the graph regression, considering that most of methods for classification on confidence calibration generally cannot be generalized to the regression, we think CaGCN may not be directly applied to the graph regression task. This could be another future work. Thanks.
>
> [1] Kipf T N, Welling M. Semi-supervised classification with graph convolutional networks[J]. arXiv preprint arXiv:1609.02907, 2016.
>
> [2] Veličković P, Cucurull G, Casanova A, et al. Graph attention networks[J]. arXiv preprint arXiv:1710.10903, 2017.
>
> [3] Song H, Diethe T, Kull M, et al. Distribution calibration for regression[C]//International Conference on Machine Learning. PMLR, 2019: 5897-5906.

---

### Decision · Program_Chairs · 2021-09-27

**Decision:**

Accept (Poster)

**Comment:**

This paper explores an interesting problem that lies at the intersection of two hot topics: graph neural networks and the reliability of NNs.
The observation that GNNs under-estimate themselves, as opposed to regular NNs, is also interesting, and the technical contributions to solving this problem are also solid, and thus can be expected to be of interest to the NeurIPS community.